# Uncovering the rewired IAP-JAK regulatory axis as an immune-dependent vulnerability of LKB1-mutant lung cancer

Changfa Shu [1,2], Jianfeng Li[1], Jin Rui[3], Dacheng Fan [1], Qiankun Niu[1], Ruiyang Bai[1,4], Danielle Cicka[1], Sean Doyle[1], Alafate Wahafu[1,5], Xi Zheng[1,6], Yuhong Du[1,7,8], Andrey A. Ivanov [1,7,8], Deon B. Doxie[3], Kavita M. Dhodapkar [8,9], Jennifer Carlisle[3,8], Taofeek Owonikoko[3,8], Gabriel Sica[8,10], Yuan Liu [8,11], Suresh Ramalingam [3,8], Madhav. V Dhodapkar [3,8], Wei Zhou [3,8] ✉, Xiulei Mo [1,8] ✉ & Haian Fu [1,3,7,8] ✉

Harnessing the power of immune system to treat cancer has become a core clinical approach. However, rewiring of intrinsic circuitry by genomic alterations enables tumor cells to escape immune surveillance, leading to therapeutic failure. Uncovering the molecular basis of how tumor mutations induce therapeutic resistance may guide the development of intervention approaches to advance precision immunotherapy. Here we report the identification of the Liver Kinase B1 (LKB1)-Inhibitor of Apoptosis Protein (IAP)- Janus Kinase 1 (JAK1) dynamic complex as a molecular determinant for immune response of LKB1-mut lung cancer cells. LKB1 alteration exposes a critical dependency of lung cancer cells on IAP for their immune resistance. Indeed, pharmacological inhibition of IAP re-establishes JAK1-regulated Stimulator of interferon genes (STING) expression and DNA sensing signaling, enhances cytotoxic immune cell infiltration, and augmentes immune-dependent antitumor activity in an LKB1-mutant immune-competent mouse model. Thus, IAP-JAK1-targeted strategies, like IAP inhibitors, may offer a promising therapeutic approach to restore the responsiveness of immunologically-cold LKB1-mutant tumors to immune checkpoint inhibitors or STING-directed therapies.

Impressive clinical activity of immune system-oriented anti-tumor strategy has led to the rapid rise of immunotherapy as standard cancer care[1-5]. However, the varying primary response rates and emerging acquired resistance present daunting challenges in expanding the impact of immunotherapy[6-9]. In parallel with the immune-targeted effort to search for immune checkpoint molecules and neoantigens, tumor-intrinsic factors have been suggested to modulate tumor immune-responsiveness[10-12]. For example, loss-of-function mutations

[1]Department of Pharmacology and Chemical Biology, Emory University School of Medicine, Atlanta, GA, USA. [2]Department of Obstetrics and Gynecology, The Third Xiangya Hospital, Central South University, Changsha, Hunan, P R China. [3]Department of Hematology and Medical Oncology, Emory University, Atlanta, GA, USA. [4]Department of Dermatology, Xiangya Hospital, Central South University, Changsha, China. [5]The First Affiliated Hospital, Medical School of Xi'an Jiaotong University, Xi'an, Shannxi, P R China. [6]Cancer Institute, the Second Affiliated Hospital, Zhejiang University School of Medicine, Hangzhou, Zhejiang, P R China. [7]Emory Chemical Biology Discovery Center, Emory University School of Medicine, Atlanta, GA, USA. [8]Winship Cancer Institute of Emory University, Atlanta, GA, USA. [9]Aflac Cancer and Blood Disorders Center, Children's Healthcare of Atlanta, Emory University, Atlanta, GA, USA. [10]Department of Pathology and Laboratory Medicine, Emory University School of Medicine, Atlanta, GA, USA. [11]Rollins School of Public Health, Emory University, Atlanta, GA, USA. ✉e-mail: wzhou2@emory.edu; xmo@emory.edu; hfu@emory.edu

in β2-microglobulin and Janus kinases (JAK) and amplification of Cyclin D1 have been reported in patients resistant to immunotherapy[13–15]. Therefore, understanding how oncogenic drivers determine the intricate tumor immune response will be critical for developing biomarkers and modulators to improve immunotherapy efficacy[16].

Liver kinase B1 (LKB1), also known as Serine/threonine kinase 11, is a tumor suppressor that regulates AMP-activated protein kinases[17,18]. The LKB1-SIK axis has been shown to suppress the development of non-small cell lung cancer[19,20]. LKB1 mutations (mut) frequently occur in lung adenocarcinoma (LUAD)[21,22]. LKB1-mut LUAD tumors gain cellular fitness advantage in part through mTORC1 activation, and subsequent metabolic rewiring and epigenetic reprogramming[23]. However, LKB1-mut LUAD is considered as undruggable due to the loss-of-function mutations found in tumors and is resistant to chemotherapy, targeted therapy, and immunotherapy[9,24–26].

The role of LKB1-mut in shaping the suppressive tumor-immune microenvironment is emerging[8,9,27–29]. It has been reported that LKB1-mut is a major genetic driver of primary resistance to immune checkpoint inhibitors[9,30,31]. LKB1-mut LUAD has been characterized to exhibit an immune-suppressive phenotype through multiple mechanisms. For example, LKB1-mut LUAD cells have altered expression of proinflammatory cytokines and immune checkpoint molecules, reprogrammed immune infiltration, and remodeled extracellular matrix[9,29,32–36]. It has been demonstrated that LKB1-mut LUAD is associated with repressed expression of stimulator of interferon genes (STING) and corresponding tumor-intrinsic DNA-sensing innate immune response[27,28]. These LKB1-mut associated immune-cold features have established LKB1 status as a predictive biomarker for immunotherapy response. However, the mechanisms underlying the LKB1-mut genotype and immune suppressive phenotype remain to be established. Therapeutic approaches to target LKB1-mut LUAD and reverse its immunosuppression are urgently needed.

In this work, to address these challenges, we take a focused onco-immune interactome mapping approach[37,38] to connect LKB1 with components of reported cancer immune-response pathways coupled with a chemical biology approach to examine the LKB1-mut-created dependency of tumor cells for survival. Both approaches converge at the LKB1-IAP-JAK interaction complex that couples the LKB1 status to the innate immunity regulatory function. The loss of LKB1 function in LKB1-mut cells appears to create the IAP dependency through a JAK-regulated STING innate immunity pathway. Modulators of the IAP-JAK regulatory axis, such as the IAP inhibitors, are shown to restore STING pathway function and enhance the sensitivity of LKB1-mut cells to immunotherapeutic assault.

## Results

### Onco-Immune protein-protein interaction (PPI) screening identifies cIAP1 as a LKB1 binder

LKB1-mut LUAD defines a genetic subset of lung cancer with an aggressive clinical presentation and therapeutic resistance[9,24,32]. However, the molecular connectivity bridging the LKB1-mut genotype and immune suppression phenotype remains elusive. To uncover the molecular mechanisms that determine LKB1-dictated immune response, we performed a focused oncogenic immune-regulatory protein-protein interaction (Onco-Immune PPI) mapping to link LKB1 with immune response regulatory proteins.

We constructed an Onco-Immune gene library that included 85 open-reading-frames encoding proteins with reported immune-regulatory activities, such as immunogenic cell death, antigen presentation, innate immunity, and immune checkpoint. Then, we performed an OncoImmune PPI mapping using a live cell-based BRET$^n$ (Nanoluciferase-based bioluminescence resonance energy transfer) technology to discover interactions between Nluc-tagged LKB1 and Venus-tagged Onco-Immune genes[37]. Using a stringent statistical cut-off of fold-of-change (FOC) ≥ 4.0 and $p ≤ 0.001$, we prioritized 16

positive Onco-Immune PPI hits that link LKB1 directly with multiple immune-regulatory pathways, such as the cIAP1-mediated immunogenic cell death pathway (Fig. 1A and Supplementary Data 1)[39].

To validate the LKB1-cIAP1 interaction and determine its structural basis, we performed orthogonal PPI detection assays and protein domain truncation studies. First, we confirmed that LKB1 was in complex with cIAP1 using an affinity pulldown assay (Fig. 1B) and a co-immunoprecipitation assay at the endogenous level (Fig. 1C). Similar interaction was detected between LKB1 and cIAP2, but not XIAP (Fig. 1B). Then, we conducted cIAP1 and cIAP2 domain truncation studies (Fig. 1D) and have localized the PPI interface to the N-terminal domain of baculoviral IAP repeats (BIR), particularly the BIR1-2 domain (Fig. 1E, F and Supplementary Fig. S1A, B). Reciprocally, we examined the structural basis of LKB1 for its interaction with cIAP1. The LKB1-cIAP1 interaction was primarily mediated by the LKB1 kinase domain, particularly the C-terminal lobe (the 132-347 fragment) (Fig. 1G). Interestingly, the LKB1-cIAP1 PPI was significantly reduced with LKB1 K78I, a catalytically inactive mutant (Fig. 1H). Furthermore, the naturally occurring LKB1 truncation mutants lacking the kinase domain were unable to interact with cIAP1 (Fig. 1I, J). These results suggest that the LKB1-cIAP1 PPI is LKB1 kinase-dependent. Thus cIAP1 may be dissociated from LKB1 in lung cancer cells with LKB1-inactivating mutations and therefore provide a potential vulnerability for therapeutic targeting.

### Chemical screening reveals immune response sensitizers for LKB1-mut cells

Given the largely unknown and complex oncogenic signaling underpinning LKB1-mut associated immune suppression, we carried out an unbiased high-throughput immunomodulator phenotypic (HTiP) screen to identify therapeutic vulnerability created by LKB1 mutations, particularly their vulnerability to immune-dependent therapeutic assaults.

To examine whether the established HTiP platform[40], featuring in vitro cancer- and immune-cell co-culture, could recapitulate LKB1-mut associated immune suppression, we tested the response of LUAD cells with LKB1-mut or WT, to the immune-cell attack. First, we tested a pair of isogenic cell lines: parental H1792 with LKB1-WT and the corresponding LKB1 shRNA-knockdown (KD) cells with a defined and matched genetic background. Native non-labeled human PBMCs were added to provide the allogenic immune selection pressure. We found that parental H1792 cells with LKB1-WT exhibited high sensitivity to the immune attack with significantly decreased cell viability, whereas the effect was drastically attenuated in isogenic cells with LKB1-KD (Fig. 2A). Similar results were observed in other pairs of isogenic LUAD cells, such as H1299 and H1755 cells (Supplementary Fig. S2A–C) and in a panel of patient-derived cancer cells with differential LKB1 status (Fig. 2B). These results demonstrate that the HTiP system can recapitulate the LKB1 mutation-associated intrinsic immune resistance.

To identify potential chemical probes that can restore the responsiveness of LKB1-mut cells to immune selection pressure, we performed an HTiP screen using a chemogenomic compound library with well-annotated bioactive compounds[40–43]. This screen revealed three structurally diverse Inhibitors of Apoptosis Protein (IAP) antagonists, birinapant, BV6 and GDC0152, that enhanced immune-dependent killing of LKB1-mut cells (Fig. 2C, Supplementary Fig. S2B, and Supplementary Data 2). These results reveal IAP as an intrinsic barrier for immune response. To further examine the specific impact of targeting IAP for immunomodulation, we tested the effect of three additional IAP inhibitors (IAPi), AT406, AZD5582, and LCL161. All six IAPi induced immune cell-dependent selective killing with potency in the nanomolar range (Fig. 2D). Similar immune dependency on IAP was observed in additional LKB1-mut LUAD cell lines, such as A549, H157, and H460 (Fig. 2E). Importantly, birinapant, an IAPi, enhanced the

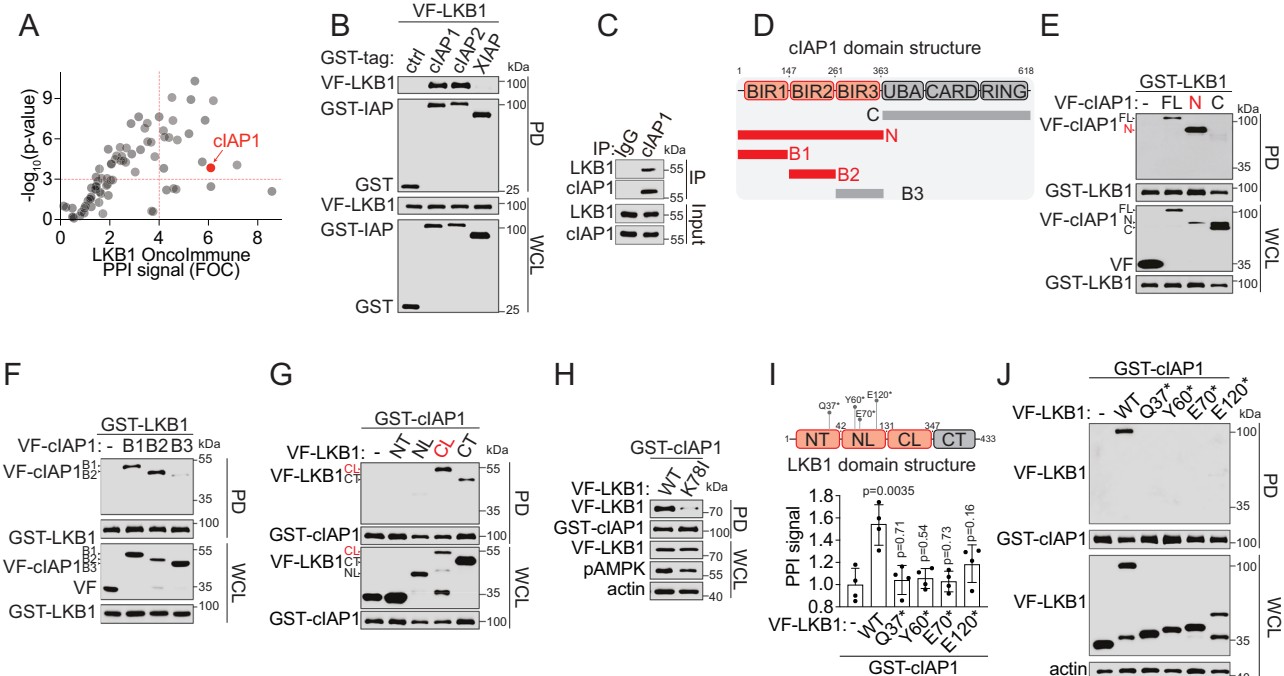

**Fig. 1 | Onco-Immune PPI profiling reveals the rewired LKB1-cIAP1 axis.**
**A** Scatter plot showing the identification of LKB1-interacting immune-regulatory protein binding partners. **B** Immunoblot showing GST-PD confirmation of LKB1-cIAP1 PPI. Cell lysate from HEK293T cells co-expressing GST-cIAP1, cIAP2, or XIAP, VF-tagged LKB1 were subjected to the GST-PD as indicated. **C** Immunoblot showing endogenous interaction of LKB1-cIAP1 with the co-IP assay in H1299 cells.
**D** Schematic illustration of the design of cIAP1 domain truncations.
**E, F** Immunoblot showing mapping of LKB1-binding domain on cIAP1. Cell lysate from HEK293T cells co-expressing GST-LKB1 and VF-tagged cIAP1 full-length (FL), N-terminal truncation (N), C-terminal truncation (C) or BIR domain truncations (B1, B2 and B3) were subjected to the GST-PD as indicated. **G** Immunoblot showing the mapping of cIAP1-binding domain on LKB1. Cell lysate from HEK293T cells co-expressing GST-cIAP1 and VF-tagged LKB1 N-terminal truncation (NT,), kinase domain N-lobe truncation (NL), kinase domain C-lobe truncation (CL), and C-terminal truncation (CT) were subjected to the GST-PD as indicated.

**H** Immunoblot showing the LKB1 kinase-dependency of LKB1-cIAP1 PPI. Cell lysate from HEK293T cells co-expressing GST-cIAP1 and VF-tagged LKB1 WT or K78I kinase-dead mutant were subjected to the GST-PD as indicated. Before collecting lysate, cells were serum-starved for overnight. **I** Schematic illustration of the LKB1 domain structures and mutations (upper) and a bar graph showing the PPI signal between cIAP1 and LKB1 WT or patient-derived mutants from TR-FRET assay (lower). The PPI signal was expressed as fold-of-change of the TR-FRET signal over the empty vector control (-) and presented as mean±SD from $n = 4$ independent experiments. *P*-values were calculated by unpaired Student's t-test with two-tailed analysis without adjustments comparing with control. **J** Representative immunoblot showing PPI signal between cIAP1 and LKB1 WT or naturally occurring mutant. Cell lysate from HEK293T cells co-expressing GST-cIAP1 and VF-tagged LKB1 WT or mutant as indicated were subjected to the GST-pulldown assay. Source data are provided as a Source Data file. For (**B**, **C**, **E**–**H**, and **J**), data are presented as one representative blot of $n = 3$ independent experiments.

response of LKB1-mut LUAD cells to both isolated CD8+ T cells and CD56+ NK cells (Fig. 2F). These results reveal the IAP status as a potential immune-dependent vulnerability in LKB1-mut LUAD cells, and targeting IAP may enhance the responsiveness of LKB1-mut tumors to immune-mediated killing. The immune-dependent anti-tumor activity of IAPi was more profound towards LKB1-KD cells than that towards the LKB1-WT counterparts (Supplementary Fig. S2A–C), which were already sensitive to immune killing, suggesting IAP as a potential tumor-intrinsic factor that controls immune sensitivity of tumors with mutated LKB1.

## IAPi restores tumor intrinsic STING expression in LKB1-mut cells
The results from both the unbiased Onco-Immune PPI mapping (Fig. 1A) and chemical screening (Fig. 2C) converge on IAP as a LKB1 binder and a therapeutic barrier in LKB1-mut LUAD cells. This physical and functional connectivity between LKB1 and IAP allowed us to use IAP inhibitors as powerful chemical tools to probe the molecular mechanisms of the immune suppressive phenotype associated with LKB1 loss. Using IAPi, we examined the biological consequence of the rewired LKB1-IAP PPI, with a focus on probing the downstream effectors of this aberrant PPI for its immune response activity.

First, we examined the effect of IAPi on the expression of known factors involved in LKB1-mut tumor-immune responsiveness,

including STING, IL-1α, IL-6, G-CSF, and PD-L1, (Fig. 2G)[9,28,29,32]. Given the potent immune-dependent anti-cancer activity of IAPi, we reasoned that the downstream effectors may also oscillate in a similar immune-dependent manner upon IAPi treatment. From a focused gene expression profiling of these factors, we found that the STING mRNA level was significantly increased upon birinapant treatment only in the presence of immune cells (Fig. 2G), whereas the mRNA levels of other factors, such as IL-1a, IL-6, GM-CSF, and PD-L1, were not significantly changed or in an immune-cell independent manner (Fig. 2G and Supplementary Fig. S3).

Suppression of tumor intrinsic STING expression was previously correlated with LKB1 genetic status and the immune resistance associated with LKB1-mut (Supplementary Fig. S4)[27,28]. To test whether STING contributes to IAPi-enhanced immune responsiveness, we performed chemical and genetic perturbation studies to examine the role of STING in birinapant-induced immune-dependent anti-tumor activity. A pharmacological loss-of-function study showed that H-151, a specific palmitoylation inhibitor of STING[44], led to a significant attenuation of birinapant-induced immune-dependent killing activity in LKB1-mut cells (Fig. 2H). Consistently, a genetic loss-of-function study via shRNA knockdown of tumor cell-intrinsic STING in LKB1-mut cells significantly blunted IAPi-induced immune-dependent killing activity

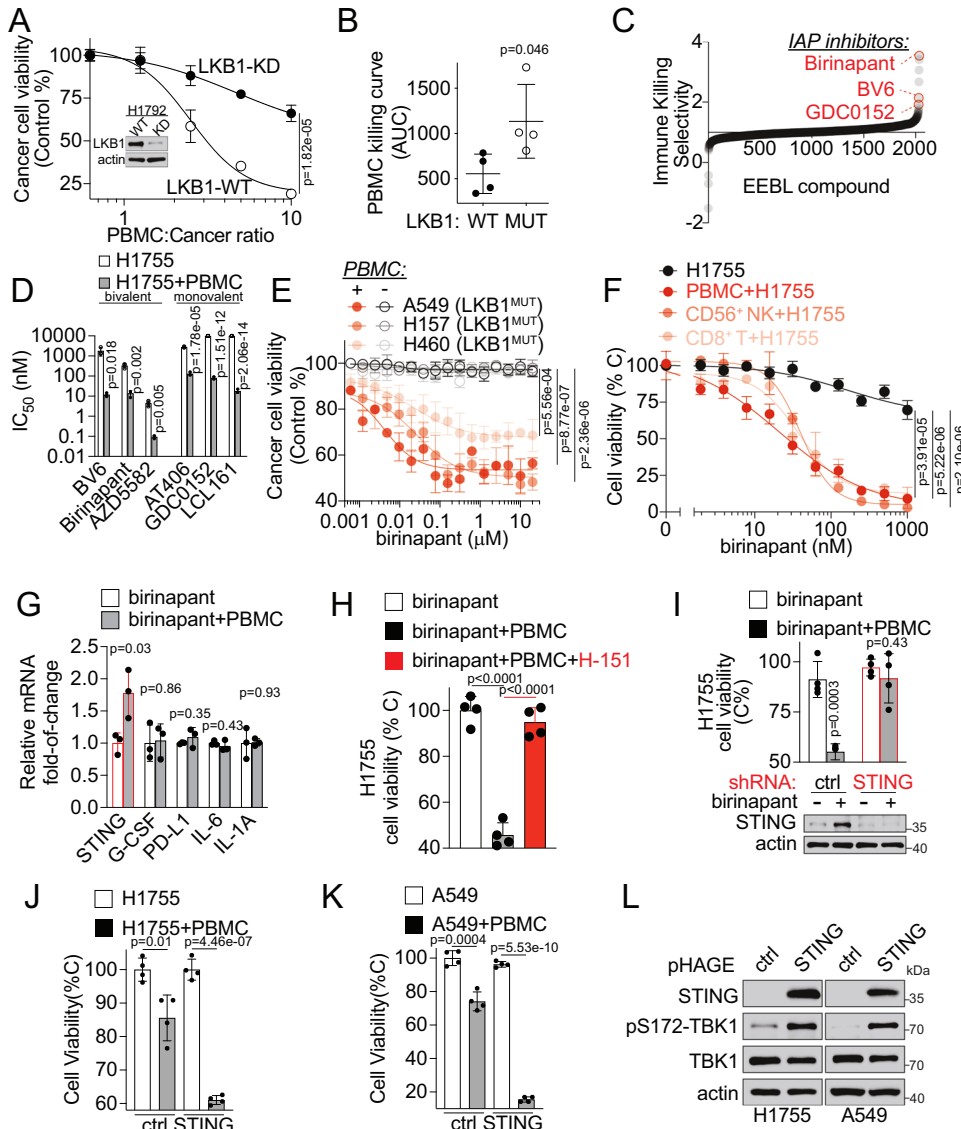

**Fig. 2 | Discovery of IAP inhibitors as immune response sensitizers in LKB1-mut lung cancer and identification of IAP-STING axis as downstream effectors of LKB1-cIAP1 PPI. A** Dose-response curve showing cell viability of isogenic LKB1-WT or KD H1792 cells co-cultured with PBMC. **B** AUC analysis of PBMC-dose dependent killing curves of lung cancer cells with LKB1-WT (Calu-1, H1299, H1792, and H292) or MUT (A549, H1792, H23 and H460). Each dot represents an individual cell line, and the data were presented as mean ± SD. **C** Selectivity of compounds in the immune cell-dependent killing of H1755 from the primary screening. **D** IC$_{50}$s of six IAP inhibitors in H1755 cancer cell alone culture versus co-culture with PBMC. **E** Dose-response confirmation of birinapant-induced immune-dependent killing in additional LKB1-mut LUAD cell lines as indicated. **F** Dose-response curves of birinapant-induced CD8$^+$ T and CD56$^+$ NK cell-dependent killing in H1755 cell. **G** STING gene expression in H1755 cells cultured alone or co-cultured with immune cells. The relative mRNA expression was expressed as fold-of-change upon birinapant (50 nM) treatment over normalized DMSO control. **H** The viability of H1755 cells

cultured alone or co-cultured with PBMC, or treated with birinapant (100 nM) or in combination with H151 (5 µM). **I** T cell viability of isogenic H1755 cells expressing non-targeting control (ctrl) shRNA or STING-targeting (STING) shRNA cultured alone or co-cultured with PBMC, or treated with birinapant (100 nM) as indicated. Immunoblot (lower) showing birinapant-induced STING expression in control H1755 cells, but not STING knockdown isogenic cells. **J, K** Cell viability of stable isogenic H1755 (J) and A549 (K) cells overexpressing STING cultured alone or co-cultured with PBMC. **L** Immunoblot showing indicated proteins of stable isogenic H1755 and A549 cells overexpressing STING by lentiviral transduction using a pHAGE-STING plasmid. One representative blot from $n = 3$ independent experiments. Source data are provided as a Source Data file. For (**A, B**) and (**D–G**), data are presented as mean ± SD from $n = 3$ independent experiments; For H-K, data are presented as mean ± SD from $n = 4$ independent experiments. *P*-values were calculated by unpaired Student's t-test with two-tailed analysis without adjustments.

(Fig. 2I). Supporting this notion, a genetic gain-of-function study showed that overexpression of STING in LKB1-mut cells per se did not compromise cancer cells autonomous viability, but significantly sensitized cancer cell's response to PBMC-mediated immune killing (Fig. 2J–L). Together, these results suggest that STING downregulation is a tumor intrinsic and immune-dependent vulnerability of LKB1-mut lung cancer cells and IAP inhibitors enhance the immune responsiveness of

LKB1-mut tumors at least in part through the STING expression restoration.

## JAK1-STAT1 mediates the IAPi effect in LKB1-mut cells

Given that birinapant-induced STING expression is immune cell-dependent, we reasoned that there must be immune co-factors involved in the IAPi effect. To probe the potential immune co-factors, we performed transcriptomic profiling to compare the

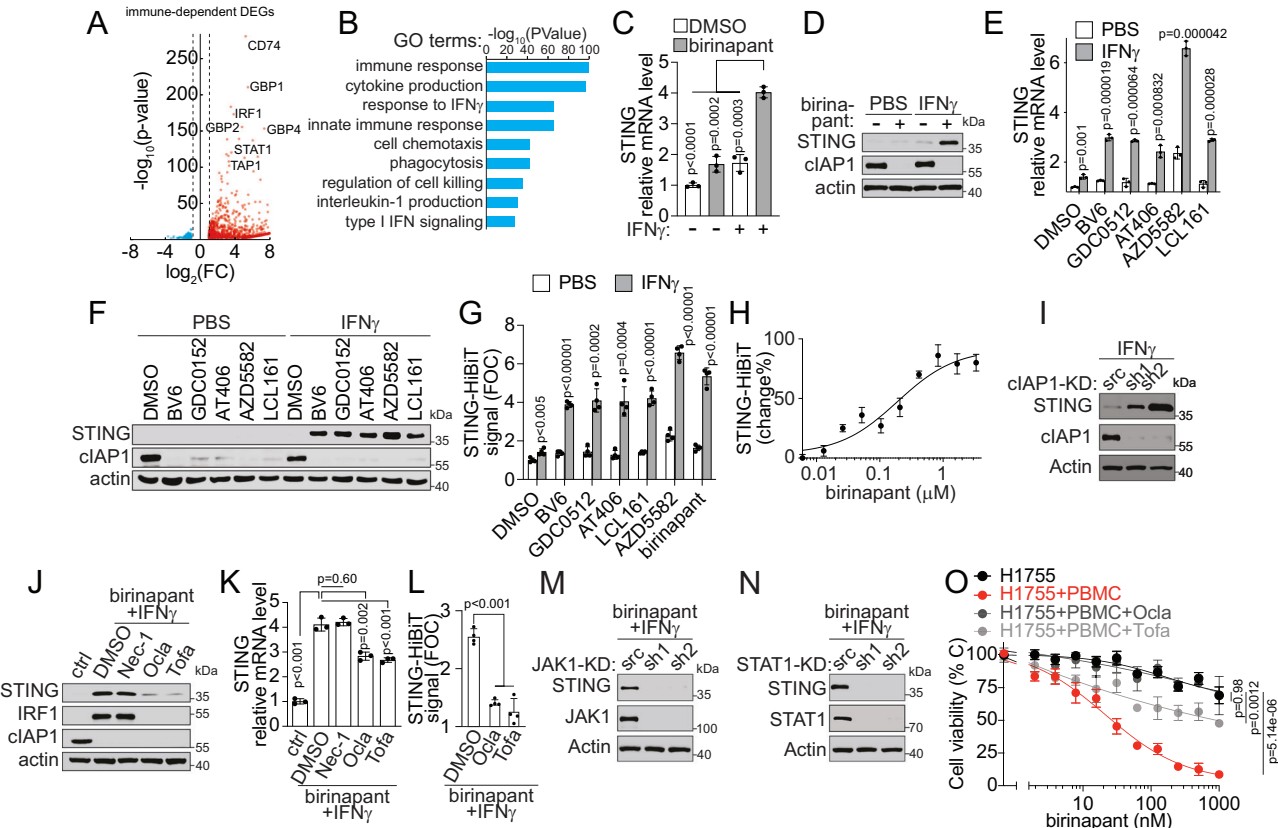

**Fig. 3 | IAP inhibitors synergize with IFNγ to induce STING expression in LKB1-mut cells. A** DEGs in H1755 cell treated with birinapant (50 nM) in cancer cells alone culture or co-cultured with PBMC (*n* = 3 technical replicates). **B** Gene ontology analysis showing DEGs-associated top-enriched pathways. **C, D** STING mRNA (C) and proteins expression (D) in A549 cells upon birinapant (500 nM) treatment without (-) or with (+) IFNγ (1 ng/mL) for 24 h. **E, F** STING mRNA (E) and protein expression (F) in A549 cells upon treatment with IAP inhibitors (500 nM) or in combination with IFNγ (1 ng/mL) for 24 h. **G** STING expression in A549 STING-HiBiT cells treated with IAP inhibitors (500 nM) or in combination with IFNγ (1 ng/mL) for 24 h (*n* = 4 independent experiments). **H** Dose-response curve of birinapant-induced STING expression in A549 STING-HiBiT cells in the presence of 1 ng/mL IFNγ. **I** STING expression in isogenic cIAP1 knockdown A549 cells treated with 1 ng/mL IFNγ for 24 h. **J, K** STING protein (J) and mRNA (K) expression in A549 cells

treated with birinapant (500 nM) and IFNγ (1 ng/mL) in combination with JAK inhibitors, oclacitinib (Ocla, 10 μM) and tofacitinib (Tofa, 10 μM), or RIPK inhibitor, necrostatin-1 (Nec-1, 10 μM). **L** STING-HiBiT signal in genetically engineered A549 cells treated with birinapant (500 nM) and IFNγ (1 ng/mL) in combination with Ocla(10 μM) and Tofa (10 μM) (*n* = 4 independent experiments). **M, N** STING expression in isogenic JAK1 (M) or STAT1 (N) knockdown (KD) A549 cells treated with birinapant (500 nM) and 1 ng/mL IFNγ for 24 h. **O** Viability of H1755 cells in cancer cell alone culture or co-culture with PBMC in combination with Ocla(10 μM) and Tofa (10 μM). Source data are provided as a Source Data file. For (**C, E, H, K,** and **O**), data are presented as mean ± SD of *n* = 3 independent experiments. For (**D, F, I, J** and **M, N**), data are presented as one representative blot of *n* = 3 independent experiments. *P*-values were calculated by unpaired Student's t-test with two-tailed analysis without adjustments.

---

differential expression genes (DEG) from LKB1-mut tumor cells in the absence or presence of immune cells upon birinapant treatment (Fig. 3A). From the DEG and gene ontology analysis, 1054 immune-dependent genes were upregulated by birinapant (Supplementary Data 3). These DEGs were enriched in several tumor immune response pathways (Fig. 3B). For example, birinapant activated Type I IFN response signaling (Fig. 3B), a reported STING-downstream event[45,46], which was suppressed in LKB1-mut LUAD (Supplementary Fig. S5A, B)[27,28].

In addition, the interferon-gamma (IFNγ) response pathway was activated by birinapant in a similar immune-dependent manner (Fig. 3A, B), while IFNγ pathway genes were significantly downregulated in LKB1-mut LUAD samples (Supplementary Fig. S5C). Moreover, the neutralization of IFNγ, but not TNFα, significantly attenuated birinapant-induced immune-dependent anti-tumor activity (Supplementary Fig. S5D). Furthermore, we and others have demonstrated that IAPi induces IFNγ production from immune cells[40,47]. These results suggest that IFNγ might be an immune co-factor that synergizes with an IAPi for its immune-dependent STING-induction and anti-tumor activity.

To test this hypothesis, we investigated the contribution of IFNγ to birinapant-induced STING restoration. When cancer cells were cultured alone, birinapant synergized with supplemented IFNγ to induce STING mRNA and protein expression in LKB1-mut cancer cells (Fig. 3C, D and Supplementary Fig. S6A–D). The induction of STING expression was likely due to the IAP inhibition effect, as all six IAP inhibitors exhibited similar synergistic effects with IFNγ on STING (Fig. 3E, F and Supplementary Fig. S6E, F). Such a synergistic effect between IAPi and IFNγ was further confirmed at the protein level using a quantitative STING-HiBiT assay[48], which is a genetically engineered A549 cell-based reporter system for monitoring endogenous STING expression (Fig. 3G and Supplementary Fig. S7). Using the STING-HiBiT assay, we found that birinapant induced a dose-dependent increase in STING protein in LKB1-mut A549 cells, with an $EC_{50}$ of ~ 0.2 μM in the presence of IFNγ (Fig. 3H). The on-target effect of IAP inhibitor-induced STING expression was confirmed by the genetic loss-of-function perturbation of cIAP1 (Fig. 3I). These results suggest that IFNγ is an immune co-factor that synergizes with IAP inhibitors to restore STING expression.

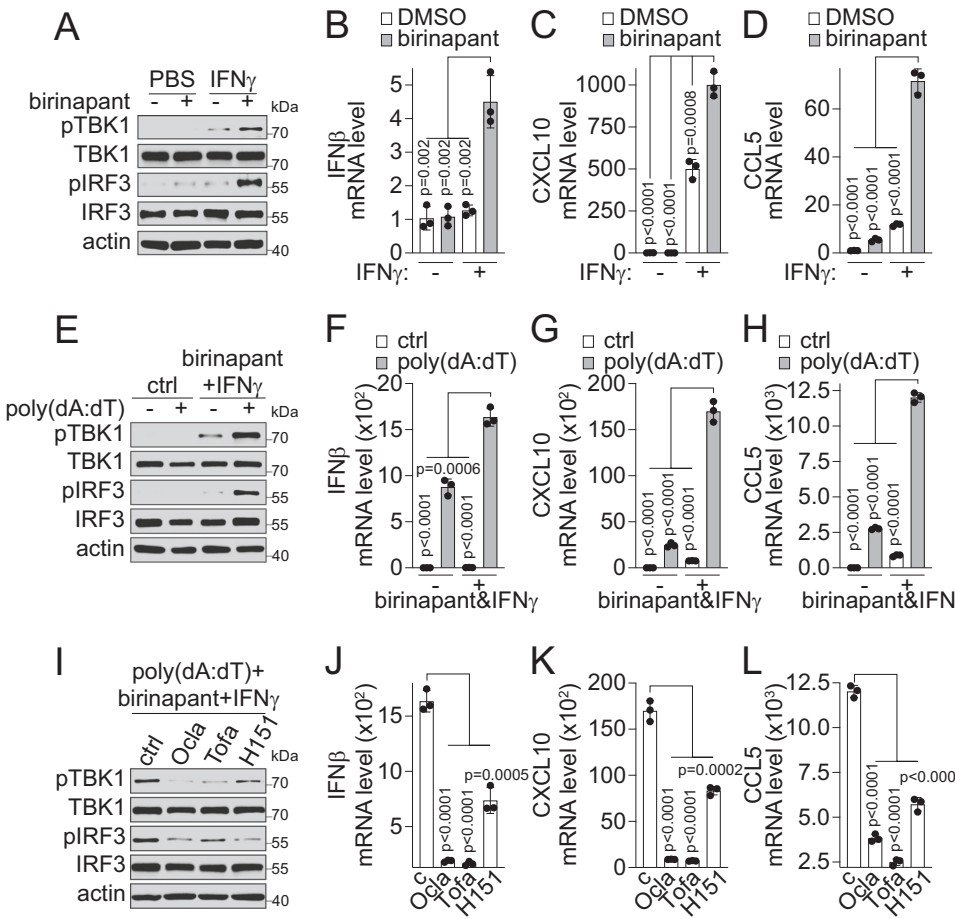

**Fig. 4 | IAP inhibitors synergize with IFNγ to induce STING-mediated DNA sensing pathway activation in LKB1-mut cells. A** Immunoblot showing indicated proteins in A549 cells treated with birinapant (500 nM) and/or IFNγ (1 ng/mL) for 24 h as indicated. **B–D** Bar graphs showing birinapant and IFNγ combination-induced expression of IFNβ (**B**), CXCL10 (**C**), and CCL5 (**D**) by qPCR in A549 cells treated with birinapant (500 nM) and/or IFNγ (1 ng/mL) for 24 h as indicated (*n* = 3 independent experiments). **E** Immunoblot showing indicated proteins in A549 cells treated with poly(dA:dT) (1 μg/mL) for 4 h in the presence or absence of 24 h pre-treatment with birinapant (500 nM) and IFNγ (1 ng/mL) combination. **F–H** Bar graphs showing poly(dA:dT)-induced expression of IFNβ (**F**), CXCL10 (**G**) and CCL5 (**H**) by qPCR in A549 cells treated with poly(dA:dT) (1 μg/mL) for 4 h in the presence or absence of 24 h pre-treatment with birinapant (500 nM) and IFNγ (1 ng/mL)

combination (*n* = 3 independent experiments). **I** Immunoblot showing indicated proteins in A549 cells treated with poly(dA:dT) (1 μg/mL) for 4 h in the presence of 24 h pre-treatment of birinapant (500 nM) plus IFNγ (1 ng/mL) in combination with Ocla (10 μM), Tofa (10 μM), or H151 (5 μM) as indicated. **J–L** Bar graphs showing JAK- and STING-dependency of birinapant-induced expression of IFNβ (**J**), CXCL10 (**K**), and CCL5 (**L**) by qPCR in A549 cells treated with poly(dA:dT) (1 μg/mL) for 4 h in the presence of 24 h pre-treatment of birinapant (500 nM) plus IFNγ (1 ng/mL) in combination with Ocla (10 μM), Tofa (10 μM) or H151 (5 μM) (*n* = 3 independent experiments). Source data are provided as a Source Data file. For (**A, E**, and **I**), data are presented as one representative blot of *n* = 3 independent experiments. For (**B–D**, **F–H**, and **J–L**), data are presented as mean values ± SD. *P*-values were calculated by unpaired Student's *t* test with two-tailed analysis without adjustments.

Given that such synergistic effects were also observed at the STING mRNA level (Fig. 3 and Supplementary Fig. S6), it is possible that a transcriptional regulatory program induced by an IAPi, or in combination with IFNγ, might be involved. To test this possibility, we examined the potential involvement of two reported effector pathways downstream of IFNγ, one pathway mediated by RIPK and the other by JAK, respectively[49,50]. We found that IAPi-induced STING restoration was significantly reduced upon treatment with JAK inhibitors, but not RIPK inhibitors, at both protein and mRNA levels (Fig. 3J, K and Supplementary Fig. S6G, H). These results were confirmed in A549 cells with the engineered STING-HiBiT reporter (Fig. 3L). In support of these results, shRNA knockdown of JAK1 and STAT1 significantly reduced the synergistic effect of IAPi and IFNγ on restoring STING expression in LKB1-mut cells (Fig. 3M, N). JAK inhibitors abolished the responsiveness of LKB1-mut tumor cells to IAPi-induced immune-dependent anti-tumor activity (Fig. 3O). These results demonstrate that IAPi-induced STING restoration and immune-

dependent anti-tumor response in LKB1-mut cells occur through the activation and sensitization of the tumor intrinsic IFNγ-JAK-STAT pathway.

## IAPi reactivates STING-mediated DNA-sensing signaling in LKB1-mut cells

Suppression of STING expression in LKB1-mut LUAD was demonstrated to lead to immune suppression via impaired cGAS-STING DNA-sensing innate immune response signaling and subsequent inhibition of the downstream TBK1-IRF3 pathway, as well as the expression of IRF3 target cytokine and chemokine genes[27,28]. Given that IAP inhibitors restored STING expression, we next examined whether IAP inhibitors could restore the STING downstream TBK1-IRF3 signaling and target gene expression.

At the basal level, without supplementation of exogenous dsDNA or cyclic dinucleotide, birinapant synergized with IFNγ to activate STING downstream signaling, as evidenced by significantly increased levels of p-TBK1 and p-IRF3 (Fig. 4A) and enhanced expression of target

cytokines and chemokine genes, such as IFNβ, CXCL10 and CCL5 (Fig. 4B–D). These results suggest that IAP inhibition not only restores STING expression but also reactivates STING-mediated DNA-sensing signaling in response to cytosolic DNA and 2′,3′-cGAMP, which could be induced by an IAP inhibitor or IFNγ treatment (Supplementary Fig. S8), or may arise from LKB1-mut tumor intrinsic genome instability[51].

Upon further STING activation using poly(dA:dT), an exogenous dsDNA, we found that the combination treatment of birinapant and IFNγ significantly augmented dsDNA-induced TBK1 and IRF3 phosphorylation (Fig. 4E) and the expression of IFNβ, CXCL10 and CCL5 (Fig. 4F–H). The augmentation effect of the TBK1-IRF3 signaling by the combination of birinapant and IFNγ was not limited to A549 cells. Similar effects were observed in additional LKB1-mut cells at the basal level (Supplementary Fig. S9A–D) or with dsDNA stimulation (Supplementary Fig. S9E–H). In addition, the birinapant and IFNγ combination treatment also significantly enhanced LKB1-mut cells' response to ADU-S100, a synthetic cyclic dinucleotide STING agonist dithio-(RP, RP)-[cyclic[A(2′,5′)pA(3′,5′)p]] (also known as ML RR-S2 CDA, MIW815, or ADU-S100)[52], as shown by increased TBK1 and IRF3 phosphorylation (Supplementary Fig. S9I). Further, we found that a JAK or STING inhibitor abolished TBK1 and IRF3 phosphorylation (Fig. 3I) and the expression of IFNβ, CXCL10, and CCL5 (Fig. 4J–L) induced by dsDNA, indicating the underlying JAK- and STING-dependency. Altogether, these results suggest that LKB1-mut cells with STING downregulation have impaired DNA-sensing signaling, which can be rescued by IAP inhibition through the IFNγ-JAK-STAT pathway-mediated STING expression.

## IAPi induces STING-mediated apoptosis of LKB1-mut cells and chemotaxis of immune cells in vitro

To explore the functional consequence of IAP inhibitor-induced STING expression and activation, we examined the fate of LKB1-mut cells and immune cell infiltration in vitro. STING activation was shown to induce apoptosis in a context-specific manner[53–56]. Given that IAP inhibitors enhanced STING expression and activation in combination with IFNγ, we sought to determine whether the combination of an IAP inhibitor with IFNγ would affect apoptosis of LKB1-mut cells. We found that birinapant alone did not lead to significant cell death, while the combination of birinapant with IFNγ induced significant caspase3/7 activation in LKB1-mut A549 cells (Fig. 5A, B). Similar effects of the birinapant and IFNγ combination on caspase3/7 activation were also observed in additional LKB1-mut cells (Supplementary Fig. S10A, B). Such a combination effect was significantly attenuated upon STING inhibition by H151 (Fig. 5C, D). These results reveal that IAP inhibition can synergize with IFNγ to induce apoptosis of LKB1-mut tumors in a STING-dependent manner.

Given that STING-mediated CXCL10 and CCL5 chemokines are involved in immune cell chemotaxis[57,58], we examined the potential functional consequence of IAP inhibitor-induced STING expression on immune cell chemotaxis in a 2D transwell assay. As shown in Fig. 5E, the birinapant treatment significantly promoted the migration of IL2- and anti-CD3-activated Jurkat T cells from the upper chamber to the cancer cell culture in the bottom chamber (Fig. 5F, G and Supplementary Fig. S10C). Similar effects of birinapant-induced immune cell migration were observed for CD56+ NK92-MI cells, or with additional dsDNA stimulation (Fig. 5F, G and Supplementary Fig. S10D–F). However, a JAK inhibitor or STING antagonist abolished the birinapant-induced immune cell migration (Fig. 5F, G and Supplementary Fig S10D–F). These results suggest that IAP inhibitor-induced STING expression and reactivation can augment immune cell migration in vitro.

## IAP inhibitor induced tumor immune response in vivo in LKB1-mut mouse models

To determine the functional consequence of targeting the rewired LKB1-cIAP1-JAK1 trimolecular complex in an in vivo setting, we examined the anti-tumor activity of IAP inhibitors using our developed LKB1-mut mouse model[36,59]. First, we found that WRJ388, a mouse tumor-derived cancer cell line isolated from our $Kras^{G12D}/Lkb1^{-/-}/p53^{WT}$ GEMM[36], has downregulated STING expression as compared to KW634 cells[60] ($Kras^{G12D}/Lkb1^{WT}/p53^{-/-}$) (Supplementary Fig. S11A, B). Moreover, IAP inhibitors in combination with mouse IFNγ significantly induced STING expression (Supplementary Fig. S11C, D) and CCL5 expression (Supplementary Fig. S11E) in $Lkb1$-mut WRJ388 cells. These results suggest that WRJ388 cells can recapitulate not only the LKB1-mut and STING-downregulation genotype-phenotype relationship observed in human LUAD patients but also the IAP dependency as shown in human LKB1-mut cells.

Using the allograft model with WRJ388 cells, we assessed the antitumor activity of birinapant in LKB1-mut background in vivo. Murine WRJ388 cells were subcutaneously injected into the immuno-competent mice, followed by birinapant treatment (Fig. 6A). Treatment with birinapant led to a ∼42% reduction in tumor volume from ∼300 to ∼175 mm³, which is ∼4.4-fold less than the vehicle control at the endpoint, indicating a significant anti-tumor effect (Fig. 6B). By contrast, no significant tumor shrinkage was observed with the birinapant treatment in the immune-deficient nude mice (Fig. 6C). Therefore, birinapant exhibits immune-dependent anti-tumor activity in vivo.

Tumor samples from the immune-competent mice were then analyzed to determine the effect of birinapant on the tumor immune microenvironment. Immunohistochemistry staining showed a significant increase in the number of tumor-infiltrating CD8+ T cells upon birinapant treatment (Fig. 6D). This increase in tumor-infiltrating CD8+ T cells was confirmed using unbiased single-cell mass cytometry profiling (Fig. 6E, F and Supplementary Data 4). Significantly, birinapant treatment restored STING expression in vivo, supporting the results from the in vitro studies (Fig. 6G). Further, the depletion of CD8+ T cells markedly reduced the antitumor efficacy of birinapant (Fig. S11F). Altogether, these results demonstrate that targeting the aberrant LKB1-cIAP1-JAK1 trimolecular complex, such as using an IAPi birinapant, restores STING expression in LKB1-mut tumors and leads to enhanced cytotoxic T cell infiltration and immune-dependent anti-tumor activity in an immune-competent mouse model.

To further examine whether IAPi could enhance the therapeutic effect of the immune checkpoint inhibitor in LKB1-mut tumors, we established a CMT167 cell-based mouse model that recapitulated the immunotherapy resistance phenotype of patients. We found that the LKB1-KO CMT167 cells showed significantly enhanced in vivo tumorigenicity capacity than the LKB1-WT control in the immune-competent mice (Supplementary Fig. S11G). Moreover, the LKB1-KO CMT167 cells showed resistance to anti-PD1 (200 μg/mouse) immunotherapy treatment (Fig. S11G, H). For potential translational studies, we selected a clinical-stage IAPi, AT406 (Fig. 2D), for the following combination test with an anti-PD1 agent. While treatment of an IAPi AT406 alone slightly slowed down the growth of the LKB1-KO CMT167 tumor as compared to the DMSO control, the combination of AT406 with anti-PD1 lead to a significant synergistic anti-tumor effect (Supplementary Fig. S11H). Further, depletion of CD8+, but not CD4+, T cells significantly abolished the anti-tumor activity of the AT406 and anti-PD1 combination treatment (Supplementary Fig. S11I), suggesting the underlying immune killing effect and CD8+ T cell-dependency.

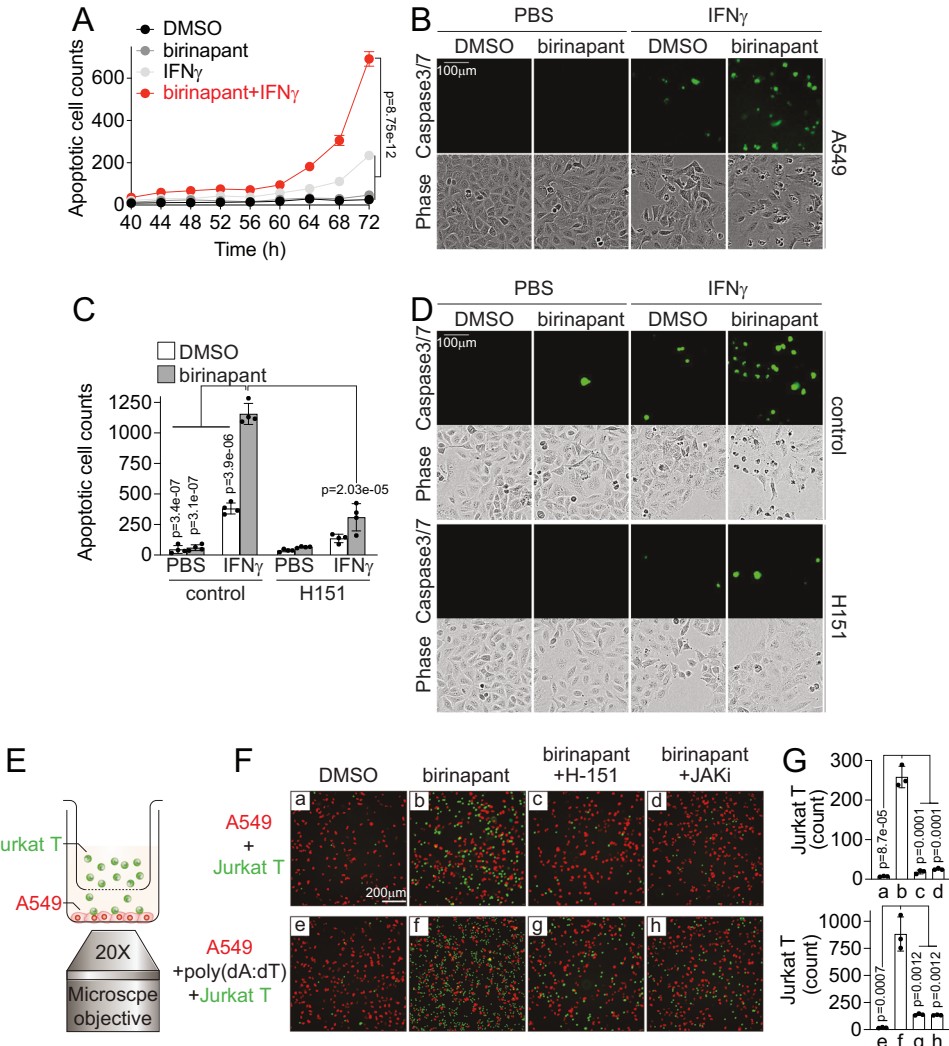

**Fig. 5 | Birinapant induces STING-mediated apoptosis of LKB1-mut cancer cells and chemotaxis of immune cells in vitro. A** Time-dependent curve of apoptotic cell counts of A549 cells treated with birinapant (500 nM), IFNγ (1 ng/mL) or in combination. ($n = 3$ independent biological replicates using different passages of A549 cells). **B** Representative images showing apoptotic A549 cells treated with birinapant (500 nM), IFNγ (1 ng/mL), or in combination. Scale bar: 100 μm. **C** Bar graph showing birinapant-induced STING-mediated cancer cell apoptosis. A549 cells were treated with birinapant (500 nM), IFNγ (1 ng/mL), H151 (5 μM,) or in combination as indicated for 72 h. ($n = 4$ independent experiments). **D** Representative images showing apoptotic A549 cells treated with birinapant (500 nM), IFNγ (1 ng/mL), H151 (5 μM), or in combination as indicated for 72 h. Scale bar: 100 μm. **E** Schematic illustration of transwell assays for measuring immune cell infiltration in vitro. **F** Representative

images showing birinapant-induced Jurkat T cells migration. A549 cells (red) and Jurkat T cells (green) were co-cultured in transwell as shown in (**E**) and were treated with birinapant (100 nM), poly(dA:dT) (1 μg/mL), or in combination with H151 (5 μM), or JAK inhibitor (JAKi), tofacitinib (10 μM), for 48 h, labeled as the letters a-h. A549 was labeled with Nuclight Red fluorescence protein, and Jurkats were pre-labeled with CellTracker™ Green CMFDA Dye. IL2 and anti-CD3 antibody were used to activate Jurkat T cells. Scale bar: 200 μm. **G** Bar graph showing the quantification of infiltrated Jurkat T cells in transwell-based migration assays. The letters a-h in lowercase were corresponding to the conditions of F ($n = 3$ independent experiments). Source data are provided as a Source Data file. For (**A**, **C**, and **G**), data are presented as mean ± SD. $P$-values were calculated by unpaired Student's $t$ test with two-tailed analysis without adjustments.

## LKB1 status controls the cIAP1-JAK1 interaction and the JAK1-STING signaling

The molecular and cellular functional studies, coupled with in vivo data, revealed the IAP-STING regulatory axis as a potential immune-dependent vulnerability created by the aberrant LKB1-cIAP1 PPI in LKB1-mut cells. The action of IAPi appears to be associated with the IFNγ-JAK-STAT signaling pathway. Therefore, we next examined the potential molecular mechanism that bridges the LKB1-IAP PPI to the JAK-STAT-STING axis with a cIAP1-directed PPI profiling assay.

From the binary interaction studies between cIAP1 and the IFNγ-JAK-STAT pathway proteins with a robust TR-FRET PPI assay[38,61–63], we found that cIAP1 strongly interacted with LKB1 and JAK1, respectively (Fig. 7A and Supplementary Fig. S12A), suggesting an intricate

interplay among LKB1, cIAP1, and JAK1. The binary cIAP1-JAK1 PPI was further confirmed by a GST-beads affinity pulldown assay with the overexpressed proteins (Fig. 7B) and a co-immunoprecipitation assay with the endogenous proteins (Fig. 7C). These results support the physical interaction of JAK1 with cIAP1 under physiologically relevant conditions.

To understand the structural basis of the interaction, truncation studies were performed with cIAP1 domains (Fig. 1D). We found that JAK1 was associated with the N-terminal domain of baculoviral IAP repeats (BIR), particularly the BIR1-2 domain (Fig. 7D, E), which is the same binding domain as LKB1 (Fig. 1E, F). These results imply that LKB1 and JAK1 may compete for interaction with cIAP1. In support of this notion, the level of JAK1 was decreased in the cIAP1 complex upon LKB1

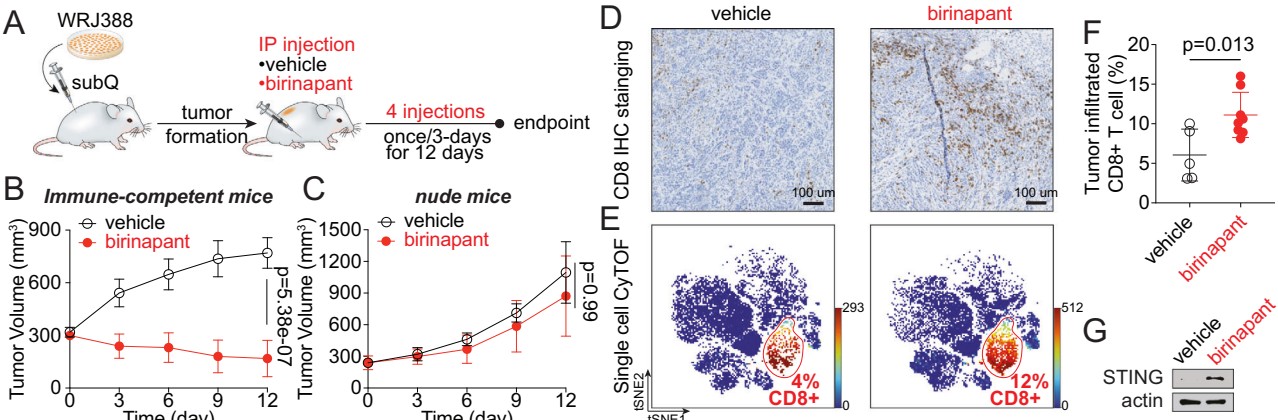

**Fig. 6 | Birinapant exhibits immune-dependent anti-tumor activity in vivo in a LKB1-mut syngeneic allograft mouse model. A** Schematics of mouse study design. **B, C** Time course of tumor volume change in immune-competent mice (**B**, *n* = 6 mice per group) and immune-deficient nude mice (**C**, *n* = 5 mice per group) treated with vehicle control or birinapant (10 mg/kg) as indicated. The data are presented as mean ± SD of the entire experimental cohort. **D** Representative images from immunohistochemistry (IHC) staining showing 2-doses birinapant-induced increase of tumor infiltrated CD8[+] T cells in vivo. Tumor samples were collected on Day 6 for the IHC staining analysis. Scale bar: 100 μm. **E** tSNE plots from single-cell mass cytometry (CyTOF) showing increased tumor infiltrated CD8[+] T cells upon 2-doses birinapant treatment in vivo. (**F**) Quantification of tumor-infiltrated CD8[+] T cells from single-cell CyTOF profiling. The data are expressed as the percentage of CD8[+] T cells in the live cell population. Each data point represents individual samples from immune-competent mice. (*n* = 5 mice for vehicle group, *n* = 7 mice for birinapant group). **G** Immunoblot showing birinapant-induced STING expression in vivo. Tumor samples harvested from the immune-competent mice at the endpoint were analyzed by SDS-PAGE and western blot with indicated antibodies. One representative blot of *n* = 3 biological replicates. Source data are provided as a Source Data file. For (**B**, **C**, and **F**), data are presented as mean values ± SD. *P*-values were calculated by unpaired Student's *t* test with two-tailed analysis without adjustments.

overexpression (Fig. 7F). These results confirmed the cIAP1-JAK1 interaction, which appears to be mutually exclusive with the LKB1-cIAP1 interaction. A similar interaction mode was detected between cIAP2 and JAK1 (Supplementary Fig. S12B–D).

Due to the competitive nature of LKB1 and JAK1 for cIAP1 interaction, the state of LKB1 interaction with cIAP1 may affect how cIAP1 regulates JAK1 and the IAP dependency of cells. To test this, we utilized both genetic and pharmacological perturbation approaches to examine the effect of the LKB1 and IAP1 status on JAK1 and its signaling. First, genetic knockdown of LKB1 led to a significant decrease in JAK1 protein, but not mRNA, suggesting a loss of JAK1 protein stability and function (Fig. 7G, H). Supporting this result, the knockdown of LKB1 impaired IFNγ signaling response, as shown with reduced STAT1 phosphorylation (Fig. 7G). Conversely, while overexpression of LKB1-WT sensitized IFNγ signaling, as measured by the STAT transcriptional reporter activity (Fig. 7I). Second, we found that birinapant, which degrades cIAP1 and thus disrupts the cIAP1-JAK1 PPI, rescued the expression of JAK1 protein, but not mRNA, in LKB1-mut cells (Fig. 7J–L and Supplementary Fig. S12E, F). This birinapant-induced increase in JAK1 protein level is likely due to the stabilization of JAK1 protein through inducing cIAP1 autodegradation and disrupting cIAP1-JAK1 PPI (Fig. 7L and Supplementary Fig. S12G). Furthermore, the upregulation of JAK1 with the birinapant treatment was correlated with significant downstream signaling activation, as shown by increased IFNγ-induced STAT1/3 phosphorylation (Fig. 7M and Supplementary Fig. S12H), STAT transcriptional reporter activity (Supplementary Fig. S12I, J), and the expression of target genes IRF1 and TAP1 (Fig. 7N-O and Supplementary Fig. S12K, L). Further, we found that cIAP1 can be tyrosine phosphorylated by JAK1, while treatment with a JAK1 inhibitor, ruxolitinib, decreased cIAP1-JAK1 PPI and cIAP1 phosphorylation (Supplementary Fig. S12M). These results suggest that lung cancer cells with LKB1 mutations have rewired cIAP1-JAK1 PPI, leading to impaired IFNγ-JAK-STAT signaling.

These data suggest that in LKB1-WT cells, LKB1 sequesters cIAP1 from the JAK1 complex, maintaining functional IFNγ-JAK-STAT signaling and the basal level of STING. Loss of LKB1 in LKB1-mut cells permits cIAP1 to bind and degrade JAK1, impairing the JAK-STAT signaling, and

leading to STING downregulation. This IAP-induced JAK-STING pathway silencing can be pharmacologically reversed by small molecule IAP inhibitors.

## Discussion

Rewiring of tumor intrinsic circuitry enables cancer cell avoidance from immune attacks, ultimately leading to therapeutic failure[10–12]. For example, lung adenocarcinomas harboring mutations in *LKB1* are highly aggressive, treatment-refractory, and insensitive to immune checkpoint inhibitors[9,24–26], representing a major clinical challenge. It is urgent to develop targeting strategies to enhance tumor response to therapies. Altogether, the discovery of rewired LKB1-cIAP1 OncoImmune PPI (Fig. 1), and the identification of IAP inhibitors as small molecule immune response sensitizers in LKB1-mut LUAD cells (Fig. 2), convergently point to a hypothesis that LKB1-mut tumors with aberrant LKB1-cIAP1 PPI may develop IAP-dependency that can be exploited as therapeutic target in an immune-dependent manner. This LKB1-mediated OncoImmune PPI network further expanded our previously identified LKB1 interactome landscape and suggested LKB1 as a major oncogenic PPI hub protein[38].

Previous studies have identified several potential therapeutic vulnerabilities, such as HSP90, in LKB1 mutant lung cancer using cancer cell line models, while the complex immune microenvironment remained to be determined due to the use of cancer cell alone culture[28,32,64–66]. Our studies with the OncoImmune PPI and HTiP approaches uncovered IAP and IAP-regulated JAK1-STING innate immunity pathway as an immune-dependent vulnerability in LKB1 mutated lung cancer cells for overcoming the immunotherapy resistance. Mechanistic examination revealed an intricate dynamic protein interaction hub by which LKB1 interacts with IAP in competition with JAK1, which offers a working model that the loss of LKB1 in LKB1-mut cancer cells allows IAP to engage JAK1 to control JAK1-mediated IFNγ response and the effector STING pathway, underpinning the IAP dependency. In support of this model, pharmacological inhibition of IAP with birinapant synergizes with IFNγ to reverse STING expression and function to promote tumor cytotoxic T cell infiltration and induce apoptotic death of LKB1-mut lung cancer cells. Importantly, such an

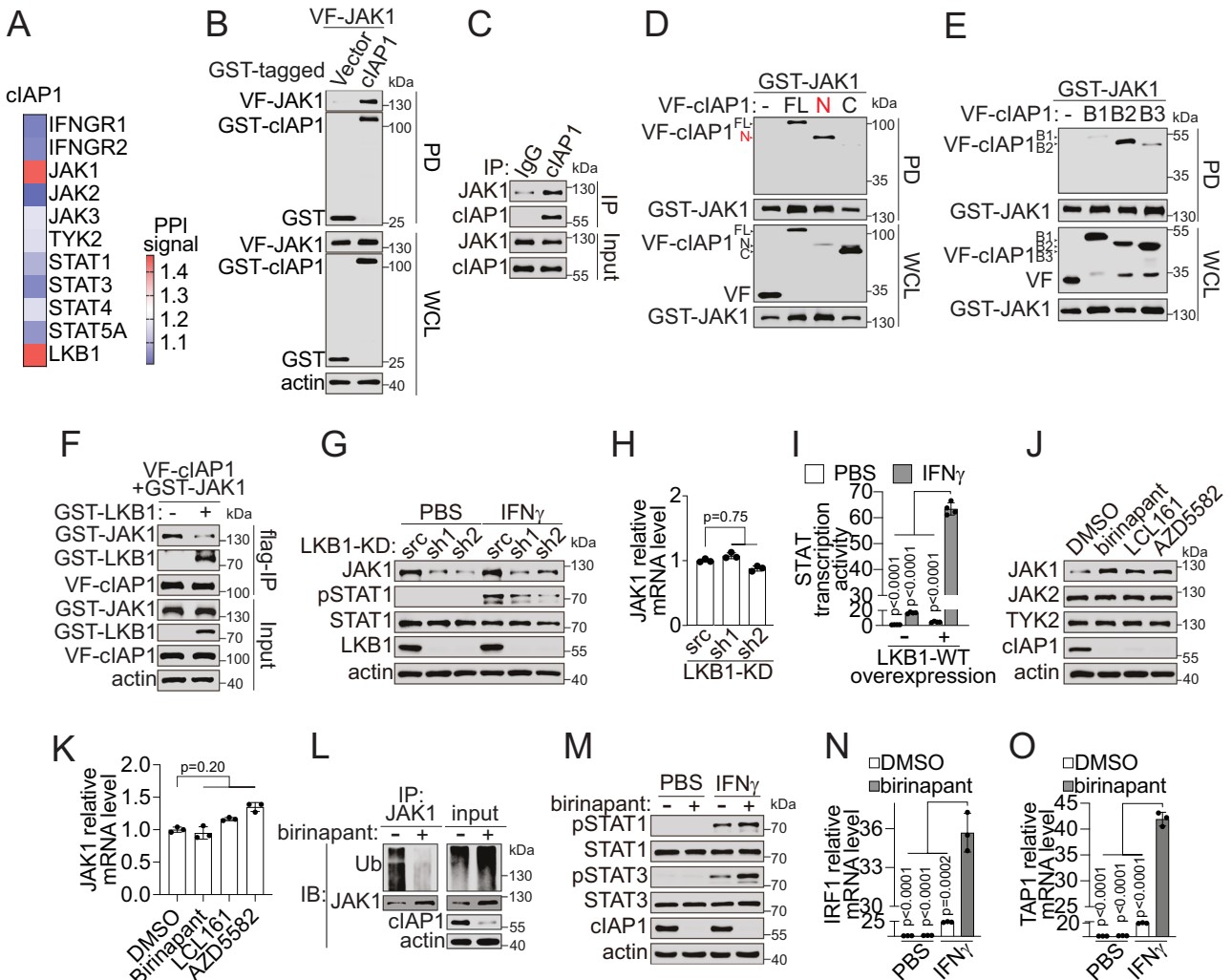

**Fig. 7 | Identification of the LKB1-cIAP1-JAK1 trimolecular complex in shaping IAP- and STING-dependency in LKB1-mut cells. A** TR-FRET PPI signal between cIAP1 and IFNγ-JAK-STAT pathway proteins. Data were presented as the average of $n = 3$ independent experiments. **B** GST-PD confirmation of cIAP1-JAK1 PPI in HEK293T cells. **C** Endogenous interaction of cIAP1-JAK1 with the co-IP assay in A549 cells. **D, E** Mapping of JAK1-binding domain on cIAP1 in HEK293T cells co-expressing GST-JAK1 with and VF-tagged cIAP1 full-length (FL), N-terminal truncation (N), C-terminal truncation (C) or BIR domain truncations (B1, B2 and B3). **F** Competitive binding between LKB1 and JAK1 with cIAP1. **G, H** Immunoblot showing indicated proteins (**G**) and qPCR showing JAK1 mRNA expression (**H**) ($n = 3$ independent experiments) in isogenic LKB1-KD H1299 cells treated with 1 ng/mL IFNγ or PBS for 1 h. **I** STAT-driven ISRE transcriptional activity in HEK293T cells transfected with ISRE-luc reporter plasmid with (+) or without (−) LKB1-WT

overexpression, and treated with 1 ng/mL IFNγ or PBS for 24 h ($n = 4$ independent experiments). **J, K** JAK1 protein (**J**) and mRNA expression (**K**) ($n = 3$ independent experiments) upon IAP inhibitor treatment in A549 cells treated with IAP inhibitors (500 nM) as indicated for 24 h. **L** JAK1 ubiquitination in A549 cells with 18 h birinapant (500 nM) followed by 6-hour MG132 (20 μM) treatment. **M** Immunoblot showing indicated proteins in A549 cells treated with IFNγ (1 ng/mL) or PBS for 1 h with (+) or without 24 h pretreatment of birinapant (500 nM). **N, O** IRF1 (**N**) and TAP1 (**O**) mRNA expression in A549 cells treated with birinapant (500 nM), IFNγ (1 ng/mL), or in combination for 24 h ($n = 3$ independent experiments). Source data are provided as a Source Data file. For (**B–G, J**, and **L, M**), data are presented as one representative blot of $n = 3$ independent experiments. For (**H, K**, and **N, O**), data are presented as mean values ± SD. *P*-values were calculated by unpaired Student's *t* test with two-tailed analysis without adjustments.

immune-dependent mode of action of IAP inhibitors in LKB1-mut cancer cells revealed by the HTiP approach is strongly supported by in vivo data. Birinapant effectively induced shrinkage of LKB1-mutant tumors in an immune-competent mouse model while showing no effect in an immune-deficient nude mouse model, presenting a potential therapeutic strategy for lung cancer patients with mutated LKB1.

The regulatory complex uncovered in our study couples the status of IAP to the activation state of the JAK1-STING pathway. Downregulation of tumor intrinsic STING expression has been frequently observed in many tumor types[27,28,67–70]. However, the regulatory mechanisms underlying STING expression control are context-dependent and remain to be elucidated[28,71–73]. In the

context of LKB1-mut LUAD, STING expression downregulation has been associated with transcriptional repression[27,28]. Distinct from the previously identified mechanism of epigenetic reprogramming mediated by DNMT1 and EZH2[28,74], our data reveal a transcriptional program by which the IFNγ-JAK-STAT immune response signaling pathway dictates the STING expression in LKB1-mut cells. Our data further suggest that tumor intrinsic STING expression could be a potential biomarker for immune responsiveness, while restoring STING expression could be effective therapeutic strategies in LKB1-mut LUAD. Our study offers alternative approaches with the JAK1-directed mechanisms to reactivate the STING pathway in various tumors, which may be synergistic with the reported epigenetic modulation approach.

Dysregulation of IAP and JAK has been identified in a number of tumor types. For example, overexpression of IAPs has been linked to tumor progression, evasion of apoptosis, and poor prognosis[75–77]. On the other hand, the loss-of-function JAK mutations have been associated with immune evasion and primary resistance to immunotherapy[13]. In contrast to the aberrant IAP and JAK function due to alterations at the genetic level, our data suggest that cIAP1 and JAK1 may be dysregulated at the protein level via rewired protein-protein interactions. Our study suggests that sequestration of IAP1 by LKB1 helps maintain a functional IFNγ-JAK-mediated immune response in LKB1-WT cells, whereas, in LKB1-mut cells, IAP1 directly impinges on JAK1 and impairs tumor immune response. Thus, IAP's pro-oncogenic activity could be enhanced by the loss of its bound LKB1 in LKB1 mutant cancer, while JAK1 could be directly weakened by IAP interaction, leading to STING suppression and immune resistance. These results implicate that such an immune response regulatory mechanism at the JAK1 protein level might be a general phenomenon in other tumor types with WT JAK1[14]. Moreover, the rewired JAK1 PPIs, such as IAP1-JAK1 in LKB1-mut cells, might be a promising therapeutic target, inhibition of which may sensitize tumors for enhanced immune response. Like the demonstrated IAP inhibitors, the IAP/JAK1 PPI antagonists may have the potential to specifically re-activate the STING activity for therapeutic development.

The discovered IAP-JAK1 interaction suggests an emerging pathway for the action of IAP inhibitors as anticancer immune enhancers. IAP inhibitors, also known as SMAC mimetics, have been studied extensively as potential anti-tumor agents[78–83]. IAP inhibitors were designed to block the inhibitory effect of IAP on caspase-mediated apoptosis signaling and thus promote cell death. Our data suggest that IAP inhibitors alone have minimal cell death-inducing effect on LKB1-mut tumors. However, immune factors, such as IFNγ, is required for IAP inhibitors to achieve their potent anti-tumor effect. IAP inhibitors synergize with IFNγ to activate the RIPK1-dependent cell death pathway in a colon cancer cell model. In LKB1-mut LUAD tumors, however, the IAP inhibitor and IFNγ combination is JAK1-dependent involving the STING activation. Thus, the effect of the IAP inhibitor and IFNγ combination on its effector signaling could be context-dependent, adding another layer of complexity to the emerging immunomodulatory activities of IAP inhibitors[40,47,84–88].

Our results may have significant implications for the treatment of immune cold tumors. LKB1-mut LUAD has been characterized to exhibit an immune suppressive phenotype with multiple immune-cold features, such as low PD-L1 expression, low tumor mutational burden, altered proinflammatory cytokine profiles, reprogrammed immune infiltration, remodeled extracellular matrix, and recently demonstrated STING expression downregulation[9,27–29,32–36]. Therefore, effective strategies that re-inflame LKB1-mut tumors may have significant therapeutic potential. Our data suggest that IAP inhibitors may impact both immune infiltration and tumor-killing phase in part through restoring tumoral STING expression, reinvigorating STING-mediated DNA-sensing pathway, re-inducing innate immunity cytokine and chemokine production, promoting chemotaxis of immune cells, and re-sensitizing tumors to IFNγ-mediated immune responsiveness. Our data suggest that IAP inhibitors may be promising immunotherapy adjuvants with immune checkpoint inhibitors or STING agonizts to further overcome LKB1-mut-associated immune resistance.

To further determine the translational potential of our findings, future studies with long-term survival experiments in more disease-relevant GEMM models are needed. In our in vivo study, the mice in the control group reached the maximal tumor burden permitted in a short-term within two weeks. Therefore, we evaluated the in vivo efficacy of IAP inhibitors within the same short-term treatment, and collected paired tumor samples to gain mechanistic insights into the LKB1-IAP-JAK regulatory complex in LKB1 mutated tumors. However,

we recognize the importance of long-term survival studies to comprehensively evaluate the in vivo efficacy of IAP inhibitors and its combination effect with anti-PD1 immunotherapy for clinical translation. In addition, we used subcutaneous syngeneic mouse models, which may have distinct immunological profiles as compared to the orthotopic GEMM models. Future studies using additional LKB1-mut LUAD GEMM models by considering the clinically relevant co-occuring mutated genes, such as *KRAS* and *TP53*, are needed to delve deeper into the long-term effect and to determine potential treatment-associated adaptive resistance mechanisms.

## Methods

### Ethics statement
This research complies with all relevant ethical regulations. All animal studies were approved and conducted according to the Emory University Institutional Animal Care and Use Committee (IACUC) guidelines. A cumulative tumor burden scoring system was used for comprehensive assessment of the animal's condition related to tumor development, including tumor size limit of 18–20 mm, body weight, tumor ulceration, mobility and other body conditions. Human peripheral blood mononuclear cells from healthy volunteer donors were purchased from ATCC.

### In vitro cell culture
All cell lines were incubated at 37 °C in humidified conditions with 5% $CO_2$. Human embryonic kidney 293 T cells (HEK293T; ATCC, CRL-3216) were maintained in Dulbecco's Modified Eagle's Medium (DMEM; Corning, #10-013-CV). Human non-small cell lung cell lines, including A549 (ATCC, CCL-185), Calu-1 (ATCC, HTB-54), H1299 (ATCC, CRL-1848), H1755 (ATCC, CRL-5892), H1792 (ATCC, CRL-5895), H1944 (ATCC, CRL-5907), H23 (ATCC, CRL-5800), H292 (ATCC, CRL-1848), and H460 (ATCC, HTB-177), were cultured in Roswell Park Memorial Institute(RPMI) 1640 medium. The isogenic LKB1-wildtype and mutant cells were generated by lentivirus-transduction of corresponding shRNA or cDNA plasmids in the parental cells. Human immune cells, including Jurkat T cell (ATCC, TIB-152), NK92-MI (ATCC, CRL-2408), peripheral blood mononuclear cells (normal PBMC, ATCC, PCS-800-011), CD8+ T cells (Stemcell, 200-0164) and CD56+ NK cells (Stemcell, 70037) were cultured in RPMI-1640 medium. Cell culture medium was supplemented with 10% fetal bovine serum (ATLANTA biologicals, #S11550) and 100 units/ml of penicillin/streptomycin (Cell Gro, Cat# 30-002-CI). All cell lines were authenticated via short tandem repeat (STR) profiling at regular intervals to confirm their identity. We routinely tested all cell lines for mycoplasma contamination using PCR-based MycoStrip® by InvivoGen kit (Cat# rep-mysnc-100).

### Cell line and mouse model for in vivo studies
Mouse lung adenocarcinoma cells, WRJ388[59] and KW634 (generous gifts from Dr. Kwok-Kin Wong)[89] were cultured in RPMI-1640 medium. CMT167 cells (CancerTools, Cat. #: 151448) were cultured in a DMEM medium. CMT167 cells were derived from metastasizing mouse lung cancer CMT64 cells and carry KRAS^G12V mutation, and are TP53- and LKB1-WT[90–93]. CMT167 LKB1-WT scramble control and the isogenic LKB1-knockout cells were generated by lentiviral-based CRISPR/Cas9 technique with sg RNA targeting mouse LKB1 or non-target control (Applied Biological Materials, cat# 45728114 and K010). Five- to 6-week-old male athymic nude mice (18–20 g) were purchased from Harlan Laboratories. Syngeneic immune-competent *KL* (*Kras*^G12D/ *Lkb1*^-/-) female mice were generated from *Kras*^G12D*Lkb1*^fl/fl*Rosa-luc* GEMM (KL-GEMM) model[36,59]. Eight-week-old C57BL/J6 mice were purchased from The Jackson Laboratory.

### Plasmids
Plasmids for mammalian expression of Glutathione S-transferase-(GST), Nanoluc (NLuc-), Venus-Flag- (VF), and Human influenza

hemagglutinin- (HA) tagged proteins were generated using the Gateway cloning system (Invitrogen, Waltham, MA, USA) according to the manufacturer's protocol. The WT STK11 (LKB1) gene in pDONR223 was purchased from the OpenBiosystem Kinome Entry ORF set. Other genes, such as OncoImmune-related genes, in pDONR221, were purchased from Human ORFeome V8.1 set. HA-Ubiquitin was purchased from Addgene (#18712)[94]. The LKB1 and cIAP1 domain truncations in pDONR223 were generated using PCR and Gateway cloning system. The LKB1 mutations, including K78I and nonsense mutations, were introduced using QuikChange Lightning Site-Directed Mutagenesis Kit (Agilent Technologies, Cat# 210518). STING and LKB1 cDNA were cloned into pHAGE (plasmid HIV-1 Alex Gustavo George Enhanced) lentiviral vector for lentivirus packaging. Plasmids for shRNA knockdown were purchased from the MISSON shRNA library (Sigma-Aldrich, see Supplementary Data 5). All the plasmids were confirmed by sequencing.

## Transfection

Polyethylenimine(PEI, Cat#23966) transfection reagent was used for plasmid transfection in HEK293T cell. FuGene HD (Roche, Cat# E2920) was used in a ratio of 3 µl to 1 µg DNA for transfection in other cancer cells.

## Onco-Immune PPI mapping

We used BRET[n] technology in a miniaturized uHTS 1536-well plate-based format to assess PPIs between LKB1 and tumor-intrinsic immune-response pathway proteins in live cells. NLuc- and Venus-fusion proteins allow streamlined monitoring of protein expressions simultaneously with BRET signal detection in a simple add-and-read mode. Briefly, HEK293T cells (1500 cells in 4 µl per well) were cultured in 1536-well plates at 37 °C before they were co-transfected in wells with Venus-tagged OncoImmune-related genes in combination with NLuc-tagged STK11 genes using Linear polyethylenimines (PEIs, Polysciences, 23966). Transfection was performed by adding 1 µl mixture of PEI (30 ng/µl) and DNA (10 ng/µl) to 4 µl cell culture, assisted by robotic operations with the Biomek NX[P] Lab Automation Workstation (Beckman Colter). BRET saturation assay was performed by titration of DNA amount to achieve various NLuc- and Venus-tagged gene combinations.

After incubation for 48 h, Nano-Glo® luciferase substrate furimazine (Promega, N1120) was added to the cells directly. The donor luminescence signal at 460 nm and acceptor emission signal at 535 nm were measured immediately using an Envision Multilabel plate reader (PerkinElmer). The BRET[n] signal is expressed as the ratio of light intensity measured at 535 nm over that at 460 nm. The specific BRET[n] signal for the interaction of two proteins is expressed as net BRET[n], which is defined as the difference in BRET signal with the co-expression of two proteins and expression of the negative control NLuc-protein only.

The relative amount of NLuc-tagged protein expression was measured by the luminescence signal at 460 nm (L460) during the BRET[n] signal measurement in the 1536-well white plate (Corning, 3727); while the Venus acceptor protein expression was detected by the Venus fluorescence intensity (FI) with excitation at 480 nm and emission at 535 nm in 1536-well black clear-bottom plate (Corning, 3893). Cells were seeded and transfected side-by-side under the same conditions for the 1536-well white plate for BRET[n] measurement and black plate for Venus FI measurement. The ratio of relative amount of acceptor over donor protein expression (Acceptor/Donor) was defined as Venus FI/L460. This intensity ratio should be proportional to the acceptor/donor ratio.

The PPI signal was quantified by fitting BRET[n] saturation curves, including one curve for PPI and two curves for empty NLuc (ctrl1) and Venus (ctrl2) controls, with each in four replicates. The saturation curves are based on the equation

$$Y = \frac{BRET_{max} \cdot X}{BRET_{50} + X} \quad (1)$$

where Y is net BRET[n], and X is Acceptor/Donor. The negative X and Y values were excluded from the analysis. For quantitative analysis, the area under the curve (AUC) was computed as a measurement of BRET[n] signal, as AUC integrates both the amplitude and shape of the saturation curve[95,96]. The fold-of-change (FOC) was calculated as $AUC_{PPI}/AUC_{max(ctrl1, ctrl2)}$ to express the difference between PPI and empty vector controls, and statistical significance ($P_{FOC}$) was calculated using Student's t-test to estimate the likelihood that $AUC_{PPI}$ is different from the $AUC_{ctrl1\&ctrl2}$.

## Immune and tumor cell co-culture assay

PBMCs, primary CD8+ T cells or CD56+ NK cells as effector immune cells, and various human lung adenocarcinoma cells as target cells were used for co-culture assays. Tumor cells were seeded at a specific density in 384-well cell culture plate (Corning #3764). Twenty-four hours later, the effector were then thawed and co-cultured in RPMI-1640 medium with tumor cells in a dose-dependent manner for four days. CD3 monoclonal antibody (100 ng/mL, OKT3, ThermoFisher) and human recombinant interleukin-2 (10 ng/mL, PeproTech) were used as activation cocktail to activate immune cells to supply immune killing factors, such as Perforin and Granzyme B.

## Cell proliferation measurement

The co-culture assay plates were imaged using the IncuCyte® S3 Live-Cell Analysis System (Essen Biosciences). The cancer cell proliferation was monitored and characterized as the percentage of confluence using the IncuCyte® basic analysis module. Because of the size distinction between effector immune cells and target cancer cells, the area filter of > 400 µm² was used to select cancer cells that are larger in size.

## Cell viability measurement

Cell Titer Blue (Promega, G8081) was added to each well. The plates were incubated for desired time at 37 °C to allow the generation of sufficient signal within the linear range. The fluorescence intensity of each well was read using an PHERAstar FSX multi-mode plate reader (Ex 545 nm, Em 615 nm; BMG LABTECH). The cancer cell viability was determined using Cell Titer Blue assay by subtracting the background signal from the PBMC alone control.

## High-throughput immunomodulator phenotypic (HTiP) screen

The primary HTiP screen was performed[40]. Briefly, parental H1755 cells harboring LKB1 mutation were seeded in 384-well cell culture plate (2000 cells/well in 40 µl medium; Corning, Cat#3764) and co-cultured with PBMCs (1000 cells/well in 10 µl media containing the activation cocktail). The 2036 Emory Enriched Library (EEL) compounds (100 nl) were used[40–42]. Compounds were added into wells in each plate using Biomek NXP Automated Workstation (Beckman) from a compound stock plate to give the final concentration of 2 µM. A parallel screening was performed with H1755 cells alone (2000 cells/well) in 50 µl medium containing the same amount of activation cocktail in the absence of PBMCs. After 4 days of incubation, image-based cell proliferation readouts followed by biochemical-based cell viability measurements were used to examine the compound effect on cancer cell growth. The percentage of control (%C) was calculated using the equation $100X(S_{compound}-S_{blank})/(S_{positive}-S_{blank})$ (2), where $S_{positive}$ and $S_{blank}$ are the corresponding average of the cell fluorescence intensity for wells with DMSO containing PBMCs/medium only or plus cancer cells, respectively. The immune killing selectivity index was calculated using the equation %C$_{-PBMC}$/%C$_{+PBMC}$ (3).

## Quantitative real-time polymerase chain reaction (qPCR)

Total RNA was isolated from cell lysates using E.Z.N.A.® Total RNA Kit I (Omega, Cat# R6834-01) and digested with DNase I (Invitrogen, Cat# 18068-015). A total of 1 μg RNA was subjected to cDNA synthesis using SuperScript™ III First-Strand Synthesis System(Invitrogen, Cat# 18080051) followed the manufacture's instruction. Reverse transcribed cDNA or isolated cytoplasmic DNA was diluted 1:5 - 1:10 in nuclease-free water. qPCR was performed using SYBR Green Supermix (Bio-rad, Cat# 1725272) in Mastercycler® RealPlex PCR System (Eppendorf) with primers as listed in Supplementary Data 1. All the primers were ordered from Eurofins Genomics LLC. (Louisville, KY, USA). The following thermal cycling conditions were used for IFNβ: 50 °C for 2 min; 95 °C for 10 min; 40 thermal cycles (94 °C for 10 sec, 59 °C for 30 sec, 72 °C for 45 sec and 75 °C for 29 sec)[97]. For other genes, the following thermal cycling conditions were used: 95 °C for 2 min; 40 thermal cycles (95 °C for 15 sec, 60 °C for 15 sec and 72 °C for 20 sec). RNA expression was normalized to GAPDH expression. Data were conducted a comparative analysis of relative expression by $2^{-\Delta\Delta Ct}$ method. Primer information of qRT-PCR was shown in Supplementary Data 1.

## Isolation of cytoplasmic dsDNA

Cytoplasmic DNA was extracted by using mitochondrial DNA isolation kit (BioVision, Cat# K280-50) according to the modified manufacturer's instructions. Briefly, $0.5 \times 10^6$ cells were lysed with $1 \times$ cytosol extraction buffer, homogenized by dounce tissue grinder(50–70 times), and then the nuclei and mitochondrial fractions were removed by centrifugation according to the manufacturer's instructions. Cytoplasmic DNA from lysate was isolated by QIAamp DNA Mini Kit(QIAGEN, Cat# 51304). Isolated cytoplasmic DNA was purified by RNaseA (Thermo Fisher Scientific, Cat# EN0531) to remove RNA contamination. The amount of mtDNA in the cytosol was determined by qRT-PCR using MT-ND1 primers. The amount of nuclear DNA in the cytosol was determined by qRT-PCR using three different sets of primers designed for different chromosomes as described previously[28]. The sequences of the primers are listed in Supplementary Data 1. 40 ng DNA per well template was used for PCR analysis as described in qRT-PCR.

## Transcriptome (RNA-seq) analysis

The birinapant-induced immune-dependent transcriptome change was analyzed by mRNA sequencing service at Novogene Corporation Inc. (Sacramento, CA,USA). Briefly, LKB1-mut cancer cells were treated by birianpant with or without PBMC co-culture for 24 hours. Then immune cells were removed by washing monolayer cancer cells three times with 1X PBS. The remaining surface-attached cancer cells were harvested for RNA sample preparation. The total RNA was isolated using E.Z.N.A.® Total RNA Kit I. RNA sequence reads were aligned to the human reference genome (GRCh38). Significantly up- or down-regulated differential expression genes (DEGs) were identified using | $\log_2$(Fold-Change (FC)) | ≥ 1 and adjusted P-value ≤ 0.05. Pathway enrichment analysis was performed using Metascape[98].

## Bioinformatics analysis

The Gene Set Enrichment Analysis (GSEA) analyses was performed as described previously[41]. Briefly, the GSEABase package in R Studio was used to score the indicating gene sets. The Hallmark gene sets available from the Molecular Signatures database (MSigDB)[99] were used as the reference gene sets. The rank of genes in indicating pathways was used in accordance with the birinapant-induced immune-dependent DEGs, or DEGs identified between LKB1-WT and LKB1-mut lung adenocarcinoma patient samples from the Genomic Data Commons (GDC) Data Portal (https://portal.gdc.cancer.gov/). The normalized enrichment score (NES) was calculated to reflect the degree in which a set of genes is overrepresented at the extremes (top or bottom) among the entire ranked list. All GSEA analyses were performed strictly according to the instructions (https://www.bioconductor.org/packages/release/bioc/vignettes/GSEABase/inst/doc/GSEABase.pdf). As for statistical significance, |NES | > 1 with a P-value and False discovery rate (FDR) < 0.05 was considered as significantly enriched.

## Cell lysate-based affinity pulldown assay

Cell lysate-based affinity-pulldown assay was performed as we previously described[38,41,61,62,100–104]. Briefly, cell lysate were prepared in nonidet P-40 (NP-40) lysis buffer (200 μL contains 1% NP-40 (IGEPAL CA-630, Sigma-Aldrich), 20 mM Tris-HCl, 150 mM NaCl, 5% glycerol and 2 mM EDTA, supplemented with protease inhibitor cocktail (Sigma-Aldrich, Cat# P8340), phosphatase inhibitor cocktail 2 (Sigma-Aldrich, Cat# P5726), and phosphatase cocktail 3 (Sigma-Aldrich, Cat# P0044)). Cell lysate were subjected to affinity-based pulldown using glutathione-conjugated sepharose beads (GE, Cat# 17-0756-05) for GST-pulldown, EZview Red Anti-Flag M2 Affinity Gel (Sigma-Aldrich, Cat# F2426) for flag-immunoprecipitation, or EZview Red Anti-HA Affinity Gel (Sigma-Aldrich, Cat# E6779) for HA-immunoprecipitation. Cell lysate were incubated with beads at 4 °C for 2 h. Beads were washed with NP-40 lysis buffer for three times. Pulldown or immunoprecipitated protein complex were eluted by boiling the beads at 95 °C for 5 min in 2x Laemmli buffer (Bio-rad, Cat# 1610737, supplemented with 200 mM DL-Dithiothreitol (DTT)). Samples were then analyzed by SDS-polyacrylamide gel electrophoresis and immunoblotting with desired antibodies.

## Co-immunoprecipitation (Co-IP) with endogenous proteins

The endogenous co-IP assay was performed as described previously[41,62,100]. Briefly, cell lysate were prepared in NP-40 lysis buffer. DTT (10 mM) and N-Ethylmaleimide (5 mM) were added for cell lysate used for protein ubiquitination level measurement. Cell lysates with total ~ 1.5 mg of proteins were used for immunoprecipitation by incubating with desired protein antibody or IgG control at 4 °C for 16 h. Then, protein A/D agarose beads were added to the cell lysate and antibody mixture for incubation at 4 °C for 1 h. Beads were then washed with NP-40 lysis buffer three times. Immunoprecipitated protein complex were eluted by boiling for 5 min at 95 °C in 2x Laemmli buffer without DTT. Then add DTT into the supernatant at a final concentration of 20 mM and then boil for another 5 min at 95 °C. Samples were analyzed by SDS-polyacrylamide gel electrophoresis and immunoblotting with desired antibodies.

## Time resolved-fluorescence resonance eEnergy Transfer (TR-FRET) assay

TR-FRET assay was performed as previously described[38]. The FRET buffer used throughout the assay contains 20 mM Tris-HCl, pH 7.0, 50 mM NaCl, and 0.01% NP-40. Cell lysate from HEK293T cells expressing GST-tagged and Venus-flag (VF) tagged donor and acceptor proteins were prepared in 1% NP-40 buffer. Cell lysate were serially diluted in FRET buffer and mixed with anti-GST-Terbium antibody (1:2000, Cisbio US Inc, Cat# 61GSTTLB). The plate was centrifuged at 200xg for 5 min and incubated at 4 °C for overnight. TR-FRET signals were measured using the BMG Labtech PHERAstar *FSX* reader with the HTRF optic module (excitation at 337 nm, emission A at 665 nm, emission B at 620 nm, integration start at 50 μs, integration time for 150 μs and 8 flashes per well). All FRET signals were expressed as a TR-FRET ratio: F665 nm /F620 nm x 10⁴.

## Western blot (Immunoblot)

Proteins in the SDS sample buffer were resolved by 10% SDS polyacrylamide gel electrophoresis (SDS-PAGE) and were transferred to nitrocellulose filter membranes at 100 V for 2 h at 4 °C. After blocking the membranes in 5% nonfat dry milk in 1 × TBST (20 mM Tris-base, 150 mM NaCl, and 0.05% Tween 20) for 1 h at room temperature, membranes were blotted with the indicated antibodies at 4 °C

overnight. Membranes were washed by 1×TBST for three times, 15 min each time. SuperSignal West Pico PLUS Chemiluminescent Substrate (Thermo, #34580) and Dura Extended Duration Substrate (Thermo, #34076) were used for developing membranes. The luminescence images were captured using ChemiDoc™ Touch Imaging System (Bio-Rad).

### Antibodies for western blot

The following antibodies were used for western blot: Flag-HRP (Sigma, Cat# A8592, dilution 1:4000), GST-HRP (Sigma, Cat# A7340, dilution 1:5000), GST(Cell Signaling Technology, Cat.# 2624S, dilution 1:1000), Flag(Cell Signaling Technology, Cat.# 14793, dilution 1:1000), HA (Cell Signaling Technology, Cat.# 3724S, dilution 1:1000), β-actin (Sigma, Cat# A5441, 1:5000 dilution), STING(Cell Signaling Technology, Cat.# 13647, dilution 1:1000), TBK1(Cell Signaling Technology, Cat.# 3504, dilution 1:1000), phospho-TBK1(S172) (Cell Signaling Technology, Cat.# 5483, dilution 1:1000), IRF3(Abcam, Cat.#ab68481, dilution 1:1000), phosphor-IRF3(S386)(Abcam, Cat.# 76493, dilution 1:500), LKB1(Invitrogen, Cat.# AHO1392, dilution 1:500), JAK1(Cell Signaling Technology, Cat.# 50996S, dilution 1:1000), JAK2(Cell Signaling Technology, Cat.# 3230S, dilution 1:1000), TYK2 anitbody(Cell Signaling Technology, Cat.# 14193S, dilution 1:1000),STAT1(Cell Signaling Technology, Cat.# 9172S, dilution 1:1000), phosphor-STAT1(Y701)(Cell Signaling Technology, Cat.# 7649S, dilution 1:1000), STAT3(Cell Signaling Technology, Cat.# 4904S, dilution 1:1000), phosphor-STAT3(Y705)(Cell Signaling Technology, Cat.# 9145S, dilution 1:1000), cIAP1(Cell Signaling Technology, Cat.# 7065, dilution 1:1000), cIAP2(Cell Signaling Technology, Cat.# 3130, dilution 1:1000), XIAP(Cell Signaling Technology, Cat.# 14334S, dilution 1:1000), Ubiquitin(Cell Signaling Technology, Cat.# 3933, dilution 1:1000), anti-mouse IgG-HRP (Jackson ImmunoResearch, Cat.# 115-035-003, dilution 1:5000), and anti-rabbit IgG-HRP (Jackson ImmunoResearch, Cat.# 111-035-003, dilution 1:5000).

### Generation of lentivirus

HEK293T cells ($5 \times 10^6$) were seeded onto a 6-well plate and transfected using PEI transfection reagent with 2 μg of pHAGE-puro-lentivirus-based expression vector, or shRNA or sgRNA vectors, together with 1.6 μg pCMV-dR8.91 and 0.66 μg of pCMV-VSVG[105]. Forty-eight to 72 h after transfection, the conditioned media containing lentivirus particles were collected and centrifuged at 4 °C, $2500 \times g$ for 15 min. Then media were filtered by 0.45 μm PVDF filter (Millipore, Cat# SLHV033RS) and stored at −80 °C.

### Interferon-stimulated response element (ISRE) and Interferon Gamma-Activated Sequence (GAS) luciferase reporter assay

HEK293T cells transiently expressing GAS or ISRE-driven firefly luciferase and internal control renilla luciferase or A549 cell stably expressing GAS or ISRE-driven firefly luciferase were used for the luciferase reporter assay. Cells were transfected with VF-tagged plasmids or treated with birinapant and/or IFNγ as indidated. Renilla and Firefly luciferase activities were measured by an Envision Multilabel plate reader (PerkinElmer) using and Dual-Glo luciferase kit (Promega, Cat# E2920) according to the manufacturer's instructions. The normalized luminescence was calculated as the ratio of luminescence of Firefly luciferase over the luminescence of Renilla luciferase for HEK293T cells. For stable A549 ISRE- or GAS-luciferase reporter cells, only the firefly luciferase signal was measured.

### Cytosolic double-strand DNA Staining

Cells were cultured on chambered cell culture slides (ibidi, Cat# 80826). Cells were treated with indicated treatments or vehicle control for 24 hours. Following the treatment, cells were incubated with culture media containing PicoGreen dsDNA stain (200-fold dilution) (Thermo Fisher, Cat# P7581) for 1 h. Then cells were incubated with

RPMI1640 medium (without FBS) containing 200 nM MitoTracker™ Red reagent (Thermo Fisher, Cat# M7512) for 20 min. Then, cells were fixed with 4% paraformaldehyde in PBS (Fisher scientific, Cat# 50-980-487) for 10 min. Cells were then washed twice with PBS and stained with Hoechst 33342 (Thermo Fisher Scientific, Cat# H3570) for 10 min. The image was acquired using Nikon A1R HD25 inverted confocal microscopes.

### STING-HiBiT expression assay

STING-HiBiT-engineered cells were generated by essentially following the protocol for CRISPR-mediated HiBiT tagging of endogenous proteins (Promega)[48]. STING-HiBiT expression was monitored using the Nano-Glo® HiBiT Lytic Detection System (Promega).

### Apoptosis assay

The cell apoptosis assay was performed essentially by following manufacturer's protocol using Incucyte® Caspase-3/7 Green Dye (SARTORIUS, Cat# 4440) (1:1000 dilution at final concentration of 5 mM). Data was aquired and analyzed using IncuCyte® S3 Live-Cell Analysis System.

### Transwell-based cell migration assay

Chemotaxis of immune cells was measured using a transwell-based cell migration assay essentially by adapting previously described methods[106]. Briefly, A549 Nuclight-red cells (SARTORIUS, #4491) was plated at the bottom chamber of the transwell plate. Twenty-four hours later, cells were treated with birinapant, JAK inhibitor, or STING inhibitor as indicated followed by adding Jurkat T cells or NK92-MI cells pre-labeld with CellTracker™ Green CMFDA Dye (ThermoFisher, # C2925) at the upper chamber. Forty-eight hours later, data was acquired and analyzed using the ImageXpress Micro HCS Imaging System.

### cGAMP ELISA assay

The cellular 2′,3′-cGAMP level was quantified using the 2′,3′-cGAMP ELISA kit from Cayman Chemical (#501700). Briefly, A549 cells were seeded in 10 cm plates and treated with either Birinapant 500 nM or human IFNγ 1 ng/mL. Twenty-four hour after treatment, cells lysate were prepared in PBS buffer supplemented with 1% NP-40. The cell lysate was centrifuged at $10,000 \times g$, 4 °C for 10 min, and the cleared supernatant were subjected to the analysis. The 2′,3′-cGAMP levels in the lysate were measured by following the manufacturer's instructions.

### Animal studies

The WRJ388 and CMT167 cell-based transplantation mouse model was used to evaluate the IAP inhibitor's anti-tumor efficacy in vivo similarly as described previously[36,59]. Briefly, a total of $2 \times 10^6$ exponentially growing WRJ-388 cells were transplanted into syngeneic immune-competent male KL mice through subcutaneous injection. CMT167 cells ($1 \times 10^6$) were allografted subcutaneously in C57BL/J6 mice. Male nude mice were used as immune-deficient control. Tumors were measured every 3 days, and the formula for tumor volume is (length x width x width)/2. Mice were randomly allocated into control and treatment groups. Both Vehicle control (Captisol or DMSO), IAP inhibitors (birinapant, intraperitoneal injections, 10 mg/kg once per 3 days; or AT406, intraperitoneal injections, 30 mg/kg once per 3 days), anti-PD1 (InVivoMAb anti-mouse PD1, CD279, BioXcell, intraperitoneal injections, 200 μg/mouse once per 3 days), or combination treatments began when the tumor kept growing between the 2 consecutive measuring and tumor volume reached about 100–300 mm³. Birinapant was dissolved in 12.5% Captisol (MedChemExpress) in distilled water. AT406 was dissolved in PBS. The solution was fresh diluted and sonicated before using. At the indicated endpoint, tumor samples were harvested for further analysis. For CD8⁺ T cell depletion, anti-mouse CD8a antibody or rat IgG2b isotype control (Bioxcell,

#BE0061 and #BE0090) was intraperitoneal injected at 10 mg/kg one day before birinapant treatment. All animal studies were approved and conducted according to the Emory University Institutional Animal Care and Use Committee (IACUC) guidelines. Mice were housed in a pathogen-free animal facility under a 12-hour light/dark cycle with constant ambient temperature ($22 \pm 2\,^{\circ}C$) and humidity (40–60%). They were provided ad libitum access to food and water. C57BL/6 J mice (Mus musculus) were used in this study. The mice were obtained from The Jackson Laboratory and were C57BL/6 J (000664). All mice were of 6–8 weeks old and maintained on a pure genetic background. The maximal tumor size/burden was not exceeded according to the guidelines.

## Single cell mass cytometry

Immune cell profiling was performed using single-cell mass cytometry[107–110]. Single-cell suspensions from mouse tumor samples were stained with 16-marker panels using metal conjugated antibodies according to manufacturer-suggested protocol using Mar Maxpar® Mouse Sp/LN Phenotyping Panel Kit (Fluidigm, Cat# 201306). Cells were fixed, permeabilized, and washed according to manufacturer's cell surface antigen staining protocol (Fluidigm). After antibody staining, cells were incubated with intercalator solution, washed, mixed with EQ Four Element Calibration Beads (catalog 201078), and acquired with mass cytometer (all reagents from Fluidigm). Gating and data analysis were performed with Cytobank (https://www.cytobank.org/). Intact viable cells were identified using cisplatin intercalator according to manufacturer-suggested concentrations (Fluidigm). viSNE analysis was performed with Cytobank.

## IHC analysis

Five-micron-thick paraformaldehyde-fixed OCT-embedded mouse lung sections or formalin-fixed paraffin-embedded mouse tumor lung sections were used for IHC analyses[59]. Slides were stained with primary CD8 antibody (Abcam, Cat# ab237723), and horse anti-rabbit IgG (Vector, Cat# MP-7401) was used as the secondary antibody. DAB substrate kit was used to develop IHC signals. These samples were blinded and analyzed by a lung cancer pathologist (G.L. Sica).

## Statistics and reproducibility

All analyses were performed using GraphPad Prism version7.0 (GraphPad Software, La Jolla, CA) or Microsoft Excel. The dose-dependent PBMC-induced or small molecule induced cancer cell growth inhibition curve was established using GraphPad Prism based on the Sigmoidal dose-response (variable slope) fitting. Statistical significance was assessed using student's $t$ test, or Wilcoxin test as indicated. Statistical tests with exact $p$-values derived, the number of independent biological replicates, and where applicable number of analyzed cells are provided in the figures and figure legends. $P$-values $\leq 0.05$ were considered statistically significant. No statistical method was used to predetermine the sample size. No data were excluded from the analyses, the experiments were not randomized, and the investigators were not blinded to allocation during experiments and outcome assessment.

## Reporting summary

Further information on research design is available in the Nature Portfolio Reporting Summary linked to this article.

## Data availability

The RNA sequencing data generated in this study have been deposited in the Gene Expression Omnibus (GEO) database under accession code GSE273406. The remaining data are available within the Article, Supplementary Information, or Source Data file. Source data are provided in this paper.

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

## Acknowledgements

This work was supported by the National Cancer Institute Office of Cancer Genomics' Cancer Target Discovery and Development (CTD²) network (U01CA217875 to H.F.), the NCI Emory Lung Cancer SPORE (P50CA217691 to S.R., H.F.) Career Enhancement Program (X.M., A.A.I.;

P50CA217691), the NCI Emory Lung cancer program project (P01CA257906 to H.F., S.R.), the NCI R01 (R01CA203928 to W.Z.), the NCI R37 (R37CA255459 to X.M.), the NCI R35 (R35CA197603 to M.V.D.), the Imagine, Innovate and Impact (I[3]) Funds from the Emory University School of Medicine and through the Georgia CTSA NIH award (UL1-TR002378), Winship Partner in Research Endowed Chair (to H.F.) and the Winship Invest$ Team Science award from Winship Cancer Institute (NIH 5P30CA138292). C.S. and R.B. are visiting students in the Emory University School of Medicine-Xiangya School of Medicine of Central South University student exchange program. A.W. is a visiting student in the Emory University School of Medicine-Xi'an Jiaotong University Health Science Center student exchange program. We thank all members of the Fu lab for technical support and comments and Xiuju Liu and Yijian Fan of the Zhou lab for technical assistance. We thank Rui Liu at the Department of Pediatrics, Emory University School of Medicine, for her help with RNAseq data analysis. We acknowledge Emory University Integrated Cellular Imaging Core Shared Resources.

## Author contributions

H.F. and X.M. conceptualized the project. C.S., D.F., Q.N., R.B., D.C., S.D., X.Z., Y.D., and X.M. performed in vitro molecular and cellular biology studies. C.S., J.L., R.J., W.Z., and X.M. performed in vivo animal studies. R.J., J.C., T.O., G.S., S.R., and W.Z. performed immunohistochemistry studies. D.B.D., K.M.D., and M.V.D. performed mass cytometry studies. A.W. and A.A.I. performed bioinformatics analysis. C.S., J.L., R.J., D.F., Y.D., K.M.D., J.C., T.O., S.R., G.S., Y. L., M.V.D., W.Z., X.M., and H.F. performed data analysis. C.S., W.Z., X.M., and H.F. wrote the initial manuscript. S.R., W.Z., X.M., and H.F. organized resources, funding, and project coordination and supervision. All authors reviewed and edited the manuscript.

## Competing interests

The authors declare no competing interests.
