## [Transparent Peer Review file · Nature Communications]

Uncovering the rewired IAP-JAK regulatory axis as an immune-dependent vulnerability of LKB1-mutant lung cancer

Corresponding Author: Dr Haian Fu

Version 0:

Reviewer comments:

Reviewer #1

(Remarks to the Author)

SYNOPSIS and IMPACT: The inhibitors of apoptosis (IAPs), in particular three IAPs with ubiquitin ligase activity, cIAP1, cIAP2 and XIAP, have a myriad of immune effects that are primarily mediated by the TNF cytokine superfamily of ligands and receptors and their signal transduction via classical or alternative NF-kappaB transcription factors or cell death pathways. These effects mostly explain how small-molecule antagonists of the IAPs, known as Smac mimetics, can kill cancer cells when combined with TNF-alpha or other death ligands. However, several groups have demonstrated that Smac mimetics can also kill cancer cells in some cases when combined with other cytokines such as interferons (IFNs), which are better known for their anti-viral effects. The mechanisms that explain this killing are less well understood. This manuscript by Shu et al (from the laboratory of Dr Fu at Emory) provides some novel insight into this killing by demonstrating that cIAP1 forms a physical protein-protein complex with the tumor suppressor kinase LKB1 as well as with the IFN immune mediator kinase JAK1. Interestingly, the authors show that the anticancer effects of Smac mimetics are mediated by the IFN-inducer STING in lung cancer cells. Importantly, the mutational silencing of the tumor suppressor LKB1 which then also provides resistance to cancer immunotherapy such as anti-PD1 biologics, can be overcome with the use of small-molecule IAP antagonists and STING activators or IFN cytokines, offering treatments and hope for lung cancer patients with those genetic deficiencies. Another group (Colombo et al, 2020) had previously shown the enhanced killing potential of Smac mimetics in LKB1 mutant lung cancer cells without providing a clear mechanism. This study also builds on Dr Fu's prior publication in 2019 in Cell Chemical Biology (Mo et al) in which they also used their immune cell and cancer cell co-culture system (called HTiP) to identify the immune-enhancing effects of Smac mimetics at killing KRAS-mutant colon cancer cells. This study is well done and uses various methodologies to prove the effects of IAP or their antagonists on IFN pathways, such as RNA-seq transcriptomic profiling, RNAi-mediated or chemical silencing of key IFN targets and pathways. They provide additional proof and roles for smac mimetics (six tested in total, 3 dimers and 3 monomers) such as birinapant, to mediate anticancer immune effects by regressing LKB1-mutant lung cancer, such as seen by the chemotaxis of CD8-positive T cells or CD56-positive NK cells, which is evident in immunocompetent mice and not immunodeficient mice. There are a few key issues mostly dealing with omissions of datasets or details on IAP selectivity or TNF dependency that if addressed would greatly strengthen the manuscript.

CONCERNS, CRITICISMS or CLARIFICATIONS NEEDED (not necessarily in order of importance, but rather in order of appearance in the text):

1. Page 6 and Figure 1A, please provide a list and table in supplemental data for all 85 ORFs/ biased candidates tested and their results. Please identify and discuss if other IAPs, such as cIAP2 or XIAP or livin or survivin, were included in that analysis as positive or negative controls and what were the results. In other words, is the LKB1 and JAK1 association only seen with cIAP1, or is the very similar IAP, cIAP2, also an interactor and what about more divergent IAP members. Many of the LUAD cancer cells, such as A549, may express low levels of cIAP2 so that may have to be specifically tested with clones or constructs in the BRET PPI assay for example.
2. Page 6, Figure 1A, spell out FOC acronym and be consistent with definition throughout text as both 'fold of change' and 'fold over control' appear in text (e.g. page 68), and it is unclear if they refer to the same thing.
3. Page 6, Figure 1A, could the authors please discuss if a hit rate of 19% (16 out of 85) in the biased screen for the BRET PPI assay is considered high or not stringent, or is expected for example.
4. Page 7 for example, Did the authors ever investigate if LKB1 or JAK1 phosphorylate cIAP1?
5. Page 8, Figure 2C, please provide a list and table in supplemental with all compound names and results from the waterfall plot for the 2036 'drugs' in the Emory EEL compound library. Are the three identified hit Smac mimetics, BV6, birinapant and

- GDC-0152, the only IAP antagonists in the library, or are there other Smac mimetics or other IAP antagonists with different mechanisms of action, such as embelin, bestatin or an ARTS mimick included in EEL?
6. Page 10, please provide a list and table for the 1054 genes induced by birinapant in the RNA-seq analysis when immune cells are present versus absent.
 7. Page 13, could the authors please specify what the supposed source of cytosolic DNA is? That is to say, are they referring to the exogenous addition of nucleic acids or is there an endogenous source (eg nuclear or mitochondrial release)?
 8. Page 13, line 277, please spell out ADU-S100 (eg ADU for Aduro?).
 9. Page 24, Figure 1D, could the authors please also add the UBA (ubiquitin-associated /binding) domain to the stick figure of cIAP1, between BIR3 and the CARD, and indicate if it is included or not in the N or C terminal constructs for example, as it is an important part of the RING E3 ubiquitin ligase function for that protein. Could the authors also indicate in the figure or legend the amino acid positions for the deletion constructs as they do for LKB1. For sake of complete the stick figure for LKB1 (eg kinase domain) and its deletion constructs could be included as well in figure 1.
 10. Throughout the text and figure legend, add PD acronym to 'pulldown complex', as is done for WCL.
 11. Page 27, Figure 2A, could the authors please indicate in legend what the method used to determine cancer cell viability from PBMCs, eg Incucyte phase-contrast adherence and size exclusion with or without caspase substrate cleavage, as it is not clear what the method is?
 12. Page 27, Fig 2C related and follow up, could the authors please also demonstrate if the Smac mimetic treated PBMCs are killing solely via IFN, or is TNF-alpha also making a contribution, by using cytokine-neutralizing antibodies as they did in their previous HTiP publication.
 13. Page 27, Fig 2D, could the authors please rearrange the bar graph (just in this case) to classify the structural classes of Smac mimetic into their 2 main differentiating classes, monovalent (AT406, LCL161, GDC0152) versus bivalent (BV6, birinapant, AZD5582), as that helps show that the known more-potent dimeric versions are killing better than the monomeric versions but that LCL161 (as shown by others) is a potent compound for a monomer.
 14. Page 27, Fig 2L, what does pHAGE mean or refer to?
 15. Page 31, Fig 3A, provide some physical space between DEGs and CD74.
 16. Page 37-39, please indicate in legend, what is making A549 and Jurkats red and green (eg FP or dye)?
 17. Page 40, Fig 6A, could the authors be clear and indicate if the ip injections of birinapant resulted in 4 or 5 total doses being administered over that timeframe?
 18. Page 43, could the authors please explain the rationale, and expected outcomes, for the 6-hr MG-132 treatment following birinapant treatment as typically proteasome inhibitors are given as pre-treatments to Smac mimetics and other perturbations to poison the proteasome and stabilize proteins.
 18. Page 40, Fig 6D, could the authors also clarify details for the tumor IHC with the 2 doses of birinapant, does that come from the mice from the experiment in 6A, and meaning those mice were sacrificed on Day X (please indicate)?
 19. Page 40, Fig 6E, could the authors please provide in Supplemental what the 16 marker antibodies (and targets) are that were used for the CyTOF mass spec cytometry analysis. Could the authors also please clarify if the CD8-positive T cell cluster shown in red is identified by any other markers, and if other CD8-positive populations exist within those other clusters not colored?
 20. Page 41, Fig 7A, if a similar profile existed for cIAP2 it would be helpful to show as well. It would also be helpful to show a JAK1 stick figure with kinase domain and deletion constructs like the other figure.
 21. Page 41, Fig 7L, does the Ub in IB mean that the ubiquitination pattern is seen for JAK1 immunoreactive bands or that the probing was actually done with anti-ubiquitin antibodies?
 22. Page 48, Fig S4C, could the authors please look up, and show, the values for the gene XAF1 (XIAP-associated factor 1) in the TCGA data, as this is a well-known and highly IFN-inducible gene similar to MX1 and OAS1, that is also a cIAP1 interactor and modulator, so it bears some relevance to this specific study.
 23. Page 58-59, Fig S9C and lines in legend, was does MI refer to for NK-92 cells?
 24. Pages 62-63, Fig S11D and lines 983-984. I am unclear as to what the authors mean when they say "representative immunoblot showing birinapant-induced decrease of ubiquitinated JAK1" when it looks like the blot is showing a decrease in the full-length form of JAK1 but that ubiquitinated species are equally present. Are they suggesting that the 6-hr post-treatment with MG-132 has stabilized those ubiquitinated forms but that some degradation of JAK1 has occurred beforehand.
 25. Pages 71-72, although the results section suggest the RNA-seq analysis is done on the lung cancer cells, the methods section description is less clear and it is difficult to know whether the immune cells were included or not in the transcriptomic analysis and therefore providing data as well? In other words, from which cells was the RNA isolated?
 26. Page 74, line 1219, missing an R in Forster or simply use Fluorescence for FRET if that is OK?
 27. Page 77, line 1302, is a more clear and accurate formula for tumor volume that the authors mean to say is: $(L \times W \times W) / 2$, ie missing the brackets and not times W/2?
 28. Page 78, line 1305, indicate if birinapant injection is via ip route if that is the case?
 29. Please mention and briefly discuss these 3 relevant publications in the introduction or discussion:
 - 29A. Colombo et al (2020) Activity of birinapant, a SMAC mimetic compound, alone or in combination in NSCLCs with different mutations. *Front Oncol* vol 10, which describes birinapant activity against lung cancer cells differing status for LKB1 and KRAS.
 - 29B. Hannes et al (2021) The Smac mimetic BV6 cooperates with STING to induce necroptosis in apoptosis-resistant pancreatic carcinoma cells. *Cell Death Dis* vol 12, which shows that BV6 synergizes with type 1 and 2 IFNs to kill pancreatic cells in a STING and TNF dependent manner.
 - 29C. Craver et al (2020) The SMAC mimetic LCL-161 selectively targets JAK2V617F mutant cells. *Exp Hematol Oncol* vol 9, which shows that JAK2 mutant cells are hyper-sensitive to LCL161 in the absence but not presence of TNF-alpha.

(Remarks to the Author)

In this manuscript, Shu and colleagues describe a therapeutic strategy to overcome the low immunogenicity of LKB1 mutant lung cancer resulting in resistance to treatment with immune checkpoint inhibitors. Through Onco-Immune PPI screening, they found that LKB1 protein directly binds to cIAP1 and attenuates its function to suppress JAK/STAT signaling. The loss of LKB1 function makes cIAP1 active, therefore LKB1 mutant lung cancer cells are highly sensitive to IAP1 inhibitors, promoting cancer cell death and immune cell migration. Mechanistically, particularly in the presence of immune cell-derived IFN-g signaling, dysregulation of JAK/STAT signaling following treatment with IAP inhibitors up-regulates STING expression and its downstream such as type-1 interferon signaling. This work is interesting, as it sheds light on the molecular mechanisms inducing resistance to immunotherapy in LKB1 mutant lung cancer associated with STING down-regulation, as well as effective therapeutic approaches. In my view, however, the main glaring weak point of this study is with regard to the lack of robust experimental data to convince the reader that their model as shown in Fig.7P is correct. Also, given that treatment with IAP1 inhibitors restores tumoral STING signaling and enhances immunogenicity, they could examine whether IAP1 inhibitors sensitize LKB1 mutant lung cancer to treatment with immune checkpoint inhibitors, which is a clear unmet clinical need. I think several points should be addressed before their publication in a journal such as Nature Communications. I have listed below some constructive suggestions:

1. Since the authors claim that IAP1 inhibitors are specifically effective in LKB1-mutant lung cancer according to their model, they should confirm their result by utilizing LKB1 depleted or LKB1 reconstituted isogenic lines as the control in the key experiments in each figure to support their conclusion/model. Throughout the paper, there are only a limited number of experiments using LKB1 isogenic lines, such as in Figure 2A. They change the cell lines for each experiment, such as the BRAF/LKB1 mutant line H1755, KRAS/LKB1 mutant line A549, and the NRAS/LKB1 mutant line H1299 without LKB1 wild-type control cells, which is insufficient to show LKB1 dependence and may mislead the reader into thinking that only cell lines that performed well in each experiment were used.
2. Related to comment 1, in in vivo experiments in Figure 6, they should use the LKB1 wild-type model, KW634, as a control to compare the therapeutic effects of IAP1 inhibitors and their impact on the immune microenvironment in the LKB1 mutant model, WRJ388.
3. Does the LKB1 mutant syngeneic murine model, WRJ388, show resistance to immune checkpoint blockade compared with KW634, as observed in the clinic? They should investigate whether a combination of IAP1 inhibitors and ICIs exhibits a synergistic effect to overcome ICI resistance in LKB1 mutant model. It would be better if they show the efficacy of IAP1 inhibitors is attenuated by treatment with CD8-neutralizing antibodies.
4. While it sounds plausible that the activation of tumoral STING triggers T-cell chemokine secretion, attracting immune cells, the mechanism by which the increase of tumoral STING expression promotes immune cell-mediated tumor cell death is less clear. This is intriguing considering the co-culture model in Figure 2, where cancer and immune cells are already mixed from the beginning. Although the molecular strategies that CD8-positive cells and NK cells kill cancer cells may vary, what sensitizes cancer cells to immune cell-induced death upon STING activation? Or, do the authors want to show that these immune cells are just sources of IFN-g supply and cancer cells result in STING-dependent intrinsic cell death? Please make the point clear why the up-regulation of tumoral STING causes “immune-dependent anti-tumor activity” in LKB1 mutant lung cancer cells.
5. In Figure 5E-G, Jurkat cells migrate towards tumor regions, even in the absence of IFN-g, despite the negligible upregulation of T-cell chemokines like CXCL10 by treatment with Birinapant alone in Figure 4. If they use Jurkat cells as the immune cells to supply IFN-g, cancer and immune cells are separately cultured in this system which is contrary to the co-culture system in Figure 2, and activation of Jurkat cells to secrete IFN-g remains unclear. To confirm their model, they should check the expression of IFN-g in Jurkat cells in the presence or absence of cancer cells. Probably they would be able to purify Jurkat cells after the co-culture experiment by a cell sorter using red color and examine RT-qPCR or another way.
6. STAT1 is a representative downstream of IFN-g signaling, and its expression should be increased following IFN-g treatment. Unlike STAT3, TBK1, and IRF3, whose activation is mainly regulated by phosphorylation alone, it is well-known that STAT1 is regulated at the expression level in addition to phosphorylation. Indeed, transcriptome analysis conducted by the authors also included STAT1 among the up-regulated genes following Birinapant treatment in Figure 3A. Conversely, in their western blotting, STAT1 expression did not show significant change, which is like internal control, following the treatment with Birinapant, and even in the presence of IFN- for 24 hours in Figure 7G. This discrepancy should be explained/reconciled.
7. In Figure S7, the mechanism responsible for the accumulation of cytoplasmic DNA following the administration of an IAP1 inhibitor or IFN-g remains totally unclear. If it is true, I think this phenomenon could be crucial and informative in this field, suggesting cGAS activation following IAP1 inhibitor or IFN-g administration. They should check the concentration of cytosolic 2'3' cGAMP by ELISA or another technique in the presence of IAP1 inhibitors or IFN-g to confirm their findings.

Reviewer #3

(Remarks to the Author)

NCOMMS-23-25213-T

Shu et al. “Uncovering the rewired IAP-JAK regulatory axis as an immune-dependent vulnerability of LKB1-mutant lung

cancer “

The authors use a variety of assays their group has developed over the years including screening for onco-immune protein-protein interaction (PPI), screening of chemical libraries with a high-throughput immunomodulator phenotype (HTiP) assay for chemicals that enhance killing of human tumor cells by human peripheral blood mononuclear cells to look for targets for treating LKB1 mutant lung adenocarcinomas (LUAD). Their primary preclinical model are human LUAD cell lines. They discover that cIAP1 interacts with LKB1 and that known inhibitors of cIAP1 (SMAC mimetics) block this interaction to increase the expression of STING in tumor cells and enhance immune cell killing and immune cell migration in the mixed culture in vitro assay. They also work out the domains of LKB1 responsible for the cIAP1 interaction and demonstrate that JAK1-STAT1 mediates the IAP inhibitor effect in LKB1 mutant LUAD. They emphasize that LKB1 loss of function in LUADs leads to an “IAP dependency.” In a syngeneic mouse model they show an cIAP1 inhibitor, birinapant, inhibits tumor cell growth and this is dependent on having an intact immune system. They conclude: “Onco-Immune Protein-Protein Interaction (OI-PPI) mapping reveals a rewired LKB1-IAP-JAK dynamic complex, informing IAP-dependency and vulnerability of LKB1-mutant tumors. Targeting this rewired OI-PPI induces immune-dependent suppression of LKB1-mutant tumors in preclinical in vitro and in vivo models, demonstrating its therapeutic potential to accelerate oncogenic alteration-directed precision immunotherapy”. No data are presented: on expression of the various factors they discovered in preclinical models in patient LKB1 mutant tumors; and no data are presented on the impact of immune checkpoint combined with IAP inhibitor therapy.

Comments to the authors:

This manuscript addresses a major knowledge gap – namely what is the mechanism underlying the “immunologically cold phenotype” of LKB1/STK11 mutant LUADs and how could this be therapeutically targeted. The paper is reviewed in the context of other recent multiple, high profile papers, addressing this same topic.

1. All of the experiments are technically well done a clearly presented.
2. Their discoveries of LKB1- cIAP1 interaction, and the subsequent JAK1-STAT1 pathway and STING effects are important and new. These clearly add to the field as a whole and fill in parts of the knowledge gap.
3. The discovery of the role of SMAC mimetics as potential therapeutics has immediate clinical translational relevance. This is also important given the initial high expectations for SMAC mimetics as cancer therapeutics yet their poor performance in the clinic to date.
4. As presented a major limitation of the current study is the information on the in vivo efficacy of the proposed treatment using only one mouse syngeneic model (WRJ388). First, this model derived from a genetically engineered mouse model (GEMM) by the authors (see their reference #36) is KRAS mutant/LKB1 mutant but TP53 wildtype. There are several other GEMM LUAD models that are KRAS, TP53, and LKB1 mutant used by other investigators. Given the frequent co-occurrence of KRAS, TP53 and LKB1 mutations before any clinical translation we will need to know the impact of a TP53 mutation on therapeutic targeting with SMAC mimetics. Second, as the authors describe the birinapant treatment led to a “~42% reduction in tumor volume..”. While this is statistically significant, what we are looking for to put the effort into tough clinical trials, are long term control/cures of LKB1 mutant LUADs in xenograft and syngeneic mouse models. The time period (12 days) of their in vivo experiment was very short compared to most of similar types of experiments reported in the literature which usually extend over multiple weeks. Finally, what we all want to know is the impact of available immune checkpoint blockade (ICB) added to this system. Could the combination of a SMAC mimetic plus ICB give long term control/potential cures in this syngeneic mouse model? It could be, that they do, or just as important they may not – whatever the results are we need to know. More information related to in vivo efficacy and/or the role of ICB, and on models with a TP53 mutation, would have greatly enhanced the value of this manuscript.
5. The discussion lacked two major things; any kind of “limitations of the current study” discussion; and also a discussion by the authors placing their study in the context and their view of its integration with all of the other recent papers which have identified potential therapeutic vulnerabilities in LKB1 mutant LUAD preclinical model systems (human and mouse models).
6. Their findings clearly have implications for what should be found in studies LKB1 mutant LUAD in patient tumor specimens with regard to the JAK1-STAT1, Sting pathways. Data on this from immuno-histochemical or multi-omics studies would have enhanced the value of the paper, whatever the results were.
6. Finally, I found it interesting and amusing to read the bioRxiv preprint version of this paper uploaded in 2021 (bioRxiv preprint doi: <https://doi.org/10.1101/2021.09.17.460294>). Clearly, the authors made the discovery in an entirely different order than the presented in the current version of the manuscript – they performed the chemical screen for therapeutics which led to the discovery of the SMAC mimetic effect in their in vitro co-culture system and that subsequently led them to look for the protein interaction. In many respects, I found their bioRxiv paper and what probably actually happened to be more scientific discovery based “compelling” (I could see how they made their discovery) than the current description of the course of events. My guess, is many other readers interested in this topic will note the same thing. “What happens in bioRxivs – does not stay (“hidden”) in bioRxivs.

Version 1:

Reviewer comments:

Reviewer #1

(Remarks to the Author)

The authors have satisfactorily addressed all of my concerns. In addition, they have greatly strengthened the manuscript by including additional requested information and data in the supplemental tables or in revised and new figures. This includes

new experiments that (1) prove their main conclusion using a second LKB1-mutant cell line, (2) demonstrate efficacy by combining a Smac mimetic with an anti-PD1 biologic in their model, (3) demonstrate the IAP selectivity for association with JAK1 to also include the cIAP1 similar IAP, cIAP2, but not the X-linked IAP, XIAP, and (4) the demonstrate (possibly for the first time) that JAK1 can phosphorylate the associated cIAP1 protein.

Reviewer #2

(Remarks to the Author)

The authors provided enough data to convince the comments, making the paper much more informative. Their manuscript can be accepted.

Reviewer #3

(Remarks to the Author)

NCOMMS-23-25213A

Shu et al. "Uncovering the rewired IAP-JAK regulatory axis as an immune-dependent vulnerability of LKB1-mutant lung cancer"

The authors submit a revised version of their manuscript with a 13 page rebuttal section addressing concerns of the 3 reviewers. The rebuttal includes additional experimental data and the manuscript has been edited to address the various concerns and the new added information. I am addressing my concerns as Reviewer #3 to their response to my major comment which I have included here for easy reference:

"4. "As presented a major limitation of the current study is the information on the in vivo efficacy of the proposed treatment using only one mouse syngeneic model (WRJ388). First, this model derived from a genetically engineered mouse model (GEMM) by the authors (see their reference #36) is KRAS mutant/LKB1 mutant but TP53 wildtype. There are several other GEMM LUAD models that are KRAS, TP53, and LKB1 mutant used by other investigators. Given the frequent co-occurrence of KRAS, TP53 and LKB1 mutations before any clinical translation we will need to know the impact of a TP53 mutation on therapeutic targeting with SMAC mimetics. Second, as the authors describe the birinapant treatment led to a "~42% reduction in tumor volume..". While this is statistically significant, what we are looking for to put the effort into tough clinical trials, are long term control/cures of LKB1 mutant LUADs in xenograft and syngeneic mouse models. The time period (12 days) of their in vivo experiment was very short compared to most of similar types of experiments reported in the literature which usually extend over multiple weeks. Finally, what we all want to know is the impact of available immune checkpoint blockade (ICB) added to this system. Could the combination of a SMAC mimetic plus ICB give long term control/potential cures in this syngeneic mouse model? It could be, that they do, or just as important they may not – whatever the results are we need to know. More information related to in vivo efficacy and/or the role of ICB, and on models with a TP53 mutation, would have greatly enhanced the value of this manuscript."

The major point was that the real ultimate translational value of their work was whether there were quantitatively large in vivo effects of combined targeting using SMAC mimetic and ICB. Another key point dealt with potential testing of their overall hypothesis using GEMM derived syngeneic mouse models of mutant KRAS, TP53, with and without LKB1/STK11 inactivation ("KP", "KPL" lines) that have been widely used for similar studies. Unfortunately they choose not to study this model, and also essentially "wrote off" any role of TP53 abnormalities. For purposes of this review, I will put aside any consideration of there not using a widely available model, and not determining the role of TP53. They did choose to study CMT167 mouse cells, which caused me to go to their methods and references to find out about these cells, what were their characteristics, and where did they get them. I spent a good deal of time searching their manuscript and figure legends but to know avail. At that point I went to the literature and found out about these cells (the parent line CMT64 derived in the 1970's as metastasizing mouse lung cancer (FRANKS, L.M., CARBONELL, A.W., HEMMING, V.J. & RIDDLE, P.N. (1976). Metastasizing tumours from serum-supplemented and serum-free cell lines from a C57BL mouse lung tumour. *Cancer Res.*, 36, 1049." and characterized in 1984 with information on the metastatic variants (Br. J. Cancer (1984), 49, 415-421 Heterogeneity in a spontaneous mouse lung carcinoma: selection and characterisation of stable metastatic variants M.G. Layton & L.M. Franks". It has been used in several publications from the Nemenoff lab who made the interesting observation that response to ICB was dramatically different in orthotopic vs. subcutaneous models who cited his source, and the derivation of the cell line and also confirmed that it had a KRASG12V mutation previously reported by Justilien et al. The status of TP53 and LKB1 are unknown. (Li et al. The Tumor Microenvironment Regulates Sensitivity of Murine Lung Tumors to PD-1/PD-L1 Antibody Blockade, *Cancer Immunol Res*; 5(9) September 2017, 767-777; Justilien V, Regala RP, Tseng I-C, Walsh MP, Batra J, et al. (2012) Matrix Metalloproteinase-10 Is Required for Lung Cancer Stem Cell Maintenance, Tumor Initiation and Metastatic Potential. *PLoS ONE* 7(4): e35040. doi:10.1371/journal.pone.0035040). I note the Nemenoff studies went for 3 weeks. The part was the length of time of studying the tumor responses to various treatment being very short – for the WRJ388 it is 12 days for CMT167 it is 9 days.

I am pointing these things out, because if this paper was presented to our grad student/post doc journal club, all of these issues would have come up. While a lot of fancy mechanism stuff is fine, what really counts is –"does this really work? Is it really worth pursuing clinically?" Of course, it is essential that the methods section document this key reagent and also that the literature is properly cited. In addition, while they have a "limitation of the current study" ending their discussion – it is pretty "tepid" given the real work that still needs to be done.

Response to reviewers' comments

Shu *et al*, Uncovering the rewired IAP-JAK regulatory axis as an immune-dependent vulnerability of LKB1-mutant lung cancer

Reviewer #1 (Remarks to the Author): with expertise in IAP targeting in cancer

SYNOPSIS and IMPACT: The inhibitors of apoptosis (IAPs), in particular three IAPs with ubiquitin ligase activity, cIAP1, cIAP2 and XIAP, have a myriad of immune effects that are primarily mediated by the TNF cytokine superfamily of ligands and receptors and their signal transduction via classical or alternative NF-kappaB transcription factors or cell death pathways. These effects mostly explain how small-molecule antagonists of the IAPs, known as Smac mimetics, can kill cancer cells when combined with TNF-alpha or other death ligands. However, several groups have demonstrated that Smac mimetics can also kill cancer cells in some cases when combined with other cytokines such as interferons (IFNs), which are better known for their anti-viral effects. The mechanisms that explain this killing are less well understood. This manuscript by Shu et al (from the laboratory of Dr Fu at Emory) provides some novel insight into this killing by demonstrating that cIAP1 forms a physical protein-protein complex with the tumor suppressor kinase LKB1 as well as with the IFN immune mediator kinase JAK1. Interestingly, the authors show that the anticancer effects of Smac mimetics are mediated by the IFN-inducer STING in lung cancer cells. Importantly, the mutational silencing of the tumor suppressor LKB1 which then also provides resistance to cancer immunotherapy such as anti-PD1 biologics, can be overcome with the use of small-molecule IAP antagonists and STING activators or IFN cytokines, offering treatments and hope for lung cancer patients with those genetic deficiencies. Another group (Colombo et al, 2020) had previously shown the enhanced killing potential of Smac mimetics in LKB1 mutant lung cancer cells without providing a clear mechanism. This study also builds on Dr Fu's prior publication in 2019 in Cell Chemical Biology (Mo et al) in which they also used their immune cell and cancer cell co-culture system (called HTiP) to identify the immune-enhancing effects of Smac mimetics at killing KRAS-mutant colon cancer cells. This study is well done and uses various methodologies to prove the effects of IAP or their antagonists on IFN pathways, such as RNA-seq transcriptomic profiling, RNAi-mediated or chemical silencing of key IFN targets and pathways. They provide additional proof and roles for smac mimetics (six tested in total, 3 dimers and 3 monomers) such as birinapant, to mediate anticancer immune effects by regressing LKB1-mutant lung cancer, such as seen by the chemotaxis of CD8-positive T cells or CD56-positive NK cells, which is evident in immunocompetent mice and not immunodeficient mice. There are a few key issues mostly dealing with omissions of datasets or details on IAP selectivity or TNF dependency that if addressed would greatly strengthen the manuscript.

CONCERNS, CRITICISMS or CLARIFICATIONS NEEDED (not necessarily in order of importance, but rather in order of appearance in the text):

1. “Page 6 and Figure 1A, please provide a list and table in supplemental data for all 85 ORFs/ biased candidates tested and their results. Please identify and discuss if other IAPs, such as cIAP2 or XIAP or livin or survivin, were included in that analysis as positive or negative controls and what were the results. In other words, is the LKB1 and JAK1 association only seen with cIAP1, or is the very similar IAP, cIAP2, also an interactor and what about more divergent IAP members. Many of the LUAD cancer cells, such as A549, may express low levels of cIAP2 so that may have to be specifically tested with clones or constructs in the BRET PPI assay for example.”

Response: We have included a supplemental data table (Table S1) for the list of 85 ORFs, which included cIAP1 (*BIRC2*), survivin (*BIRC5*) and XIAP. Our Oncolmmune-PPI screen revealed significant PPI signals between LKB1 and cIAP1, but not survivin and XIAP. To further test the selectivity among major IAP family members, we performed new experiments to test LKB1’s interaction with cIAP1, cIAP1 and XIAP using a GST-pulldown assay. We found that LKB1 interacts with cIAP2 and cIAP1, but not with XIAP (Fig. 1B). We carried out binding site mapping studies with cIAP2 truncations. We found that cIAP2’s N-terminal domain, particularly the BIR1 and BIR2 domains, contributed to its interactions with LKB1 (Fig. S1A-B). Similarly, JAK1 interacts with cIAP2 through the same domains (Fig. S12B-D), competing with LKB1 for binding to cIAP2 (Fig. S1E). See revised Fig. 1, Fig. S1, and S12.

2. “Page 6, Figure 1A, spell out FOC acronym and be consistent with definition throughout text as both ‘fold of change’ and ‘fold over control’ appear in text (e.g. page 68), and it is unclear if they refer to the same thing.”

Response: We have spelled out FOC as “fold-of-change” throughout the text and figures.

3. “Page 6, Figure 1A, could the authors please discuss if a hit rate of 19% (16 out of 85) in the biased screen for the BRET PPI assay is considered high or not stringent, or is expected for example.”

Response: We have added a discussion section on the hit rate of the BRET PPI screen. Briefly, the hit rate is expected, because we anticipate a high hit rate for a hub protein, such as LKB1, from a cancer-focused PPI mapping. For example, our previous OncoPPI study have revealed LKB1 as a major PPI hub with a hit rate of 32.5% from the lung cancer-focused PPI mapping (Nat Commun 2017). Altogether, our studies suggested LKB1 as a major oncogenic PPI hub protein in lung cancer cells. See page 20, line 1-3.

4. “Page 7 for example, Did the authors ever investigate if LKB1 or JAK1 phosphorylate cIAP1?”

Response: We have performed additional experiments and found that cIAP1 can be tyrosine-phosphorylated by JAK1. See the new Fig. S12M.

5. “Page 8, Figure 2C, please provide a list and table in supplemental with all compound names and results from the waterfall plot for the 2036 ‘drugs’ in the Emory EEL compound

library. Are the three identified hit Smac mimetics, BV6, birinapant and GDC-0152, the only IAP antagonists in the library, or are there other Smac mimetics or other IAP antagonists with different mechanisms of action, such as embelin, bestatin or an ARTS mimick included in EEL?”

Response: We have included a new supplementary table with all compound names and results for the waterfall plot. Our EEL library includes four compounds, BV6, birinapant, embelin and GDC-0152, which are annotated as IAP antagonists. Embelin was not identified as a hit from the screen. See *Table S2*.

6. *“Page 10, please provide a list and table for the 1054 genes induced by birinapant in the RNA-seq analysis when immune cells are present versus absent.”*

Response: We have included a new supplementary table with a list of 1054 DEGs induced by birinapant from the RNA-seq analysis. See *Table S3*.

7. *“Page 13, could the authors please specify what the supposed source of cytosolic DNA is? That is to say, are they referring to the exogenous addition of nucleic acids or is there an endogenous source (eg nuclear or mitochondrial release)?”*

Response: We have added clarification to the text. The supposed source of cytosolic DNA is likely from both endogenous nuclear and mitochondrial DNA release as shown in Fig. S8. See *page 13, line 11*.

8. *“Page 13, line 277, please spell out ADU-S100 (eg ADU for Aduro?).”*

Response: We have spelled-out the full name for ADU-S100: STING agonist dithio-(R_P, R_P)-[cyclic[A(2',5')pA(3',5')p] (also known as ML RR-S2 CDA, MIW815, or ADU-S100). See *page 13, line 20-22*.

9. *“Page 24, Figure 1D, could the authors please also add the UBA (ubiquitin-associated /binding) domain to the stick figure of cIAP1, between BIR3 and the CARD, and indicate if it is included or not in the N or C terminal constructs for example, as it is an important part of the RING E3 ubiquitin ligase function for that protein. Could the authors also indicate in the figure or legend the amino acid positions for the deletion constructs as they do for LKB1. For sake of complete the stick figure for LKB1 (eg kinase domain) and its deletion constructs could be included as well in figure 1.”*

Response: We have added the domain illustrations as suggested. See *revised Fig. 1D and 1I*.

10. *“Throughout the text and figure legend, add PD acronym to ‘pulldown complex’, as is done for WCL.”*

Response: We have added PD acronym to ‘pulldown complex’ as suggested.

11. “Page 27, Figure 2A, could the authors please indicate in legend what the method used to determine cancer cell viability from PBMCs, eg Incucyte phase-contrast adherence and size exclusion with or without caspase substrate cleavage, as it is not clear what the method is?”

Response: We have clarified the methods in the figure legend. Briefly, the cancer cell viability was determined using Cell Titer Blue assay by subtracting the background signal from the PBMC alone control. See revised figure legend on page 29, line 5-7.

12. “Page 27, Fig 2C related and follow up, could the authors please also demonstrate if the Smac mimetic treated PBMCs are killing solely via IFN, or is TNF-alpha also making a contribution, by using cytokine-neutralizing antibodies as they did in their previous HTiP publication”.

Response: We have tested IFN γ and TNF α dependency using the neutralizing antibodies in the *in vitro* co-culture assay. We found the birinapant-induced immune-killing activity was significantly diminished upon IFN γ -mAb treatment and slightly decreased by TNF α neutralization (Fig. S5D). These results suggest that IFN γ has a major role in Smac mimetic-induced immune killing. See Fig. S5D, page 11, line 8-10.

13. “Page 27, Fig 2D, could the authors please rearrange the bar graph (just in this case) to classify the structural classes of Smac mimetic into their 2 main differentiating classes, monovalent (AT406, LCL161, GDC0152) versus bivalent (BV6, birinapant, AZD5582), as that helps show that the known more-potent dimeric versions are killing better than the monomeric versions but that LCL161 (as shown by others) is a potent compound for a monomer.”

Response: We have rearranged the bar graph as recommended. Indeed, this regrouping showed that the dimeric versions showed a better efficacy than that of the monomeric compounds. See the revised Fig 2D.

14. “Page 27, Fig 2L, what does pHAGE mean or refer to?”

Response: We have added that pHAGE is a lentiviral vector backbone. pHAGE stands for plasmid HIV-1 Alex Gustavo George Enhanced. See page 68, line 12-13.

15. “Page 31, Fig 3A, provide some physical space between DEGs and CD74.”

Response: We have provided additional space between DEGs and CD74. See the revised Fig 3A.

16. “Page 37-39, please indicate in legend, what is making A549 and Jurkats red and green (eg FP or dye)?”

Response: We have clarified in the legend that A549 was labeled with Nuclight Red fluorescence protein and Jurkats were pre-labeled with CellTracker™ Green CMFDA Dye. See page 41, line 1-2.

17. “Page 40, Fig 6A, could the authors be clear and indicate if the ip injections of birinapant resulted in 4 or 5 total doses being administered over that timeframe?”

Response: We have revised Fig 6A and made it clear that a total of 4 doses were administered over the 12-day timeframe. See the revised Fig 6A.

18. “Page 43, could the authors please explain the rationale, and expected outcomes, for the 6-hr MG-132 treatment following birinapant treatment as typically proteasome inhibitors are given as pre-treatments to Smac mimetics and other perturbations to poison the proteasome and stabilize proteins.”

Response: We acknowledge that MG-132 was usually given as pre-treatments to determine if Smac-mimetics-induced IAP degradation is proteasome-dependent. In support of this notion, we found that MG-132 pretreatment indeed abolished birinapant-induced cIAP1 degradation (data not shown).

However, In Fig 7L, we aim to study whether disrupting cIAP1-JAK1 PPI using birinapant would lead to JAK1 protein stabilization. In this case, we need to first leverage birinapant-induced IAP degradation to disrupt the cIAP1-JAK1 PPI. We expected that birinapant disrupts cIAP1-JAK1 PPI through inducing cIAP1 degradation and thus release cIAP1-mediated JAK1 ubiquitination and degradation. Therefore, A549 cells were pre-treated with birinapant (18-hour, 500 nM) to first induce IAP degradation, followed by 6-hour MG132 (20 uM) treatment to prevent proteasomal degradation of JAK1. See the revised figure legend on page 46, line 18-20.

19. “Page 40, Fig 6D, could the authors also clarify details for the tumor IHC with the 2 doses of birinapant, does that come from the mice from the experiment in 6A, and meaning those mice were sacrificed on Day X (please indicate)?”

Response: We have added clarification that the tumor samples were collected on Day-6 for the IHC staining analysis in the figure legend. See page 42, line 10-11.

20. “Page 40, Fig 6E, could the authors please provide in Supplemental what the 16 marker antibodies (and targets) are that were used for the CyTOF mass spec cytometry analysis. Could the authors also please clarify if the CD8-positive T cell cluster shown in red is identified by any other markers, and if other CD8-positive populations exist within those other clusters not colored?”

Response: We have added a supplemental table including the 16 marker’s information. The CD8-positive T cell cluster shown in red was identified by the specific 168Er-labeled anti-CD8a monoclonal antibody (Clone 53-6.7), thus there is no other CD8-positive populations exist within other clusters. See Table S4.

21. *“Page 41, Fig 7A, if a similar profile existed for cIAP2 it would be helpful to show as well. It would also be helpful to show a JAK1 stick figure with kinase domain and deletion constructs like the other figure.”*

Response: We have performed additional experiments and showed that cIAP2 showed similar PPI with JAK1. We did not perform JAK1 domain truncation studies, so JAK1 domain figure was not included. See the new Figure S12B-D.

22. *“Page 41, Fig 7L, does the Ub in IB mean that the ubiquitination pattern is seen for JAK1 immunoreactive bands or that the probing was actually done with anti-ubiquitin antibodies?”*

Response: We have provided clarification that the probing was done with anti-ubiquitin antibody. See the revised figure legend on page 46, line 22-23.

23. *“Page 48, Fig S4C, could the authors please look up, and show, the values for the gene XAF1 (XIAP-associated factor 1) in the TCGA data, as this is a well-known and highly IFN-inducible gene similar to MX1 and OAS1, that is also a cIAP1 interactor and modulator, so it bears some relevance to this specific study.”*

Response: We found that XAF1 gene expression shows consistent correlation with LKB1 genetic status similar to other IFN-target genes, such as MX1 and OAS1. See the revised Fig S5C.

23. *“Page 58-59, Fig S9C and lines in legend, was does MI refer to for NK-92 cells?”*

Response: NK-92MI stands for IL-2-independent NK-92 cells engineered from transfection with the MFG-hIL2 vector and thus were denoted as NK-92MI (PMID: 10365666). See the revised figure legend on page 62, line 6-7.

24. *“Pages 62-63, Fig S11D and lines 983-984. I am unclear as to what the authors mean when they say “representative immunoblot showing birinapant-induced decrease of ubiquitinated JAK1” when it looks like the blot is showing a decrease in the full-length form of JAK1 but that ubiquitinated species are equally present. Are they suggesting that the 6-hr post-treatment with MG-132 has stabilized those ubiquitinated forms but that some degradation of JAK1 has occurred beforehand.”*

Response: Fig S11D (the current Fig S12G) shows a representative immunoblot from ubiquitin immunoprecipitation assay using anti-ubiquitin antibody. We overexpressed VF-JAK1 together with HA-tagged ubiquitin (HA-Ub). Then we immunoprecipitated all ubiquitinated proteins using anti-HA antibody. The immunoprecipitated (IP: HA) and the whole cell lysate (input) sample were analyzed using the antibodies as indicated. Similar to comment #18, A549 cells were treated with birinapant (500 nM) for 18-hour to drive IAP autoubiquitination and degradation, and thus IAP-JAK1 PPI disruption, followed by 6-hour MG132 (20 μ M) treatment to prevent proteasomal degradation of JAK1. In support of Fig. 7L, birinapant treatment induced IAP degradation, disrupted IAP-JAK PPI, and led

to JAK1 protein level increase in the input samples. In addition, we found that birinapant treatment induced a significantly decreased JAK1 ubiquitination in the HA-Ub-IP sample. Revised figure legend accordingly.

25. *“Pages 71-72, although the results section suggest the RNA-seq analysis is done on the lung cancer cells, the methods section description is less clear and it is difficult to know whether the immune cells were included or not in the transcriptomic analysis and therefore providing data as well? In other words, from which cells was the RNA isolated?”*

Response: We have added clarification that immune cells were removed by washing monolayer cancer cells three times with 1X PBS. The remaining surface-attached cancer cells were harvested for RNA sample preparation. See page 75, line 9-10.

26. *“Page 74, line 1219, missing an R in Forster or simply use Fluorescence for FRET if that is OK?”*

Response: We have simply used Fluorescence for FRET. See page 77, line 16.

27. *“Page 77, line 1302, is a more clear and accurate formula for tumor volume that the authors mean to say is: $(L \times W \times W) / 2$, ie missing the brackets and not times $W/2$?”*

Response: We have corrected the typo. See page 81, line 16.

28. *“Page 78, line 1305, indicate if birinapant injection is via ip route if that is the case?”*

Response: We have added the details of intraperitoneal injections. See page 81, line 16-20.

29. *“Please mention and briefly discuss these 3 relevant publications in the introduction or discussion:*

29A. Colombo et al (2020) Activity of birinapant, a SMAC mimetic compound, alone or in combination in NSCLCs with different mutations. Front Oncol vol 10, which describes birinapant activity against lung cancer cells differing status for LKB1 and KRAS.

29B. Hannes et al (2021) The Smac mimetic BV6 cooperates with STING to induce necroptosis in apoptosis-resistant pancreatic carcinoma cells. Cell Death Dis vol 12, which shows that BV6 synergizes with type 1 and 2 IFNs to kill pancreatic cells in a STING and TNF dependent manner.

29C. Craver et al (2020) The SMAC mimetic LCL-161 selectively targets JAK2V617F mutant cells. Exp Hematol Oncol vol 9, which shows that JAK2 mutant cells are hypersensitive to LCL161 in the absence but not presence of TNF-alpha.”

Response: References added. See page 90, line 11-18, ref. #78-80.

Reviewer #2 (Remarks to the Author): with expertise in lung cancer, LKB1, STING

In this manuscript, Shu and colleagues describe a therapeutic strategy to overcome the low immunogenicity of LKB1 mutant lung cancer resulting in resistance to treatment with immune checkpoint inhibitors. Through Onco-Immune PPI screening, they found that LKB1 protein directly binds to cIAP1 and attenuates its function to suppress JAK/STAT signaling. The loss of LKB1 function makes cIAP1 active, therefore LKB1 mutant lung cancer cells are highly sensitive to IAP1 inhibitors, promoting cancer cell death and immune cell migration. Mechanistically, particularly in the presence of immune cell-derived IFN-g signaling, dysregulation of JAK/STAT signaling following treatment with IAP inhibitors up-regulates STING expression and its downstream such as type-1 interferon signaling. This work is interesting, as it sheds light on the molecular mechanisms inducing resistance to immunotherapy in LKB1 mutant lung cancer associated with STING down-regulation, as well as effective therapeutic approaches. In my view, however, the main glaring weak point of this study is with regard to the lack of robust experimental data to convince the reader that their model as shown in Fig.7P is correct. Also, given that treatment with IAP1 inhibitors restores tumoral STING signaling and enhances immunogenicity, they could examine whether IAP1 inhibitors sensitize LKB1 mutant lung cancer to treatment with immune checkpoint inhibitors, which is a clear unmet clinical need. I think several points should be addressed before their publication in a journal such as Nature Communications. I have listed below some constructive suggestions:

1. *“Since the authors claim that IAP1 inhibitors are specifically effective in LKB1-mutant lung cancer according to their model, they should confirm their result by utilizing LKB1 depleted or LKB1 reconstituted isogenic lines as the control in the key experiments in each figure to support their conclusion/model. Throughout the paper, there are only a limited number of experiments using LKB1 isogenic lines, such as in Figure 2A. They change the cell lines for each experiment, such as the BRAF/LKB1 mutant line H1755, KRAS/LKB1 mutant line A549, and the NRAS/LKB1 mutant line H1299 without LKB1 wild-type control cells, which is insufficient to show LKB1 dependence and may mislead the reader into thinking that only cell lines that performed well in each experiment were used.”*

Response: We generated an additional pair of LKB1-knockin H1755 isogenic cells and assessed their sensitivity to IAP inhibitors using an *in vitro* co-culture assay. Our results showed that LKB1-mutant H1755 cells were more resistant to immune cell-mediated killing compared to their LKB1-WT counterparts (Fig. S2C). However, IAP inhibitor treatment significantly reversed this resistance in LKB1-mutant cells, but not in LKB1-WT cells that were already sensitive to immune cell-mediated killing. Similar results were observed in another pair of LKB1-knockdown H1792 isogenic cells (Fig. S2B). Altogether, our results suggest that IAP inhibitor can reverse the immune resistance phenotype of LKB1-mut cells through enhancing their responsiveness to immune cell-mediated killing. See revised Figure S2.

2. *“Related to comment 1, in in vivo experiments in Figure 6, they should use the LKB1 wild-type model, KW634, as a control to compare the therapeutic effects of IAP1 inhibitors and their impact on the immune microenvironment in the LKB1 mutant model, WRJ388.”*

Response: KW634 cells were derived from mice with mixed genetic background (~75% C57BL/6, ~25% FVB/n and 129SvEv) (PMID: 17676035), and thus have strong immunogenicity in either C57BL/6 or FVB mice due to unmatched genetic background. Therefore, while KW634 cells have been used in nude mice xenograft model, it is not suitable to use as a control to study tumor immunology in genetically diverse mice.

3. *“Does the LKB1 mutant syngeneic murine model, WRJ388, show resistance to immune checkpoint blockade compared with KW634, as observed in the clinic? They should investigate whether a combination of IAP1 inhibitors and ICIs exhibits a synergistic effect to overcome ICI resistance in LKB1 mutant model. It would be better if they show the efficacy of IAP1 inhibitors is attenuated by treatment with CD8-neutralizing antibodies.”*

Response: To test the combination effect of IAP inhibitors and ICIs, we first examined the immune response of WRJ388 to an ICI, *InVivoMAb* anti-mouse PD-1 (CD279, BioXcell). We found that WRJ388 cells respond well to ICI at 200 µg per mouse, which gave a narrow window for the combination study. Therefore, we have developed a CMT167 cell based syngeneic model that showed resistance to anti-PD-1 treatment at 200 µg dosage. We found that the LKB1-KO CMT167 cells showed significantly stronger *in vivo* tumorigenicity than the LKB1-WT control in the immune-competent mice (Fig. S11G). Moreover, the LKB1-KO CMT167 cells showed resistance to anti-PD1 (200 µg/mouse) immunotherapy treatment (Fig. S11G). For potential translation to clinic, we selected an IAP inhibitor that is currently in clinical trial, AT406, for this revision study. While treatment of an IAPi AT406 alone slightly slowed down the LKB1-KO CMT167 tumor growth as compared to the DMSO control, the combination of AT406 with anti-PD1 led to a significant synergistic anti-tumor effect (Fig. S11H). Further, depletion of CD8+, but not CD4+, T cells significantly abolished the anti-tumor activity of the AT406 and anti-PD1 combination treatment (Fig. S11I), suggesting the underlying immune killing effect and CD8+ T cell-dependency. See revised Fig. S11F-G, page 16 (line 19-23) and page 17 (line 1-7).

4. *“While it sounds plausible that the activation of tumoral STING triggers T-cell chemokine secretion, attracting immune cells, the mechanism by which the increase of tumoral STING expression promotes immune cell-mediated tumor cell death is less clear. This is intriguing considering the co-culture model in Figure 2, where cancer and immune cells are already mixed from the beginning. Although the molecular strategies that CD8-positive cells and NK cells kill cancer cells may vary, what sensitizes cancer cells to immune cell-induced death upon STING activation? Or, do the authors want to show that these immune cells are just sources of IFN-g supply and cancer cells result in STING-dependent intrinsic cell death? Please make the point clear why the up-regulation of tumoral STING causes “immune-dependent anti-tumor activity” in LKB1 mutant lung cancer cells.”*

Response: Our results suggest that IAP inhibitors may exhibit immune-dependent anti-tumor activity in LKB1-mut lung cancer cells through impacting two major steps of the tumor-immune cycle. First, our data suggest that IAP inhibitor can promote immune cell infiltration in part through upregulation of tumoral STING and downstream immune

attractive cytokine/chemokine expression. Second, our data also suggest that IAP inhibitor can sensitize tumor response to immune killing factors, such as IFN γ from the immune cells, through synergistic induction of tumoral cytosolic DNA, cGAS-STING pathway activation and IFN γ -induced cell death. See *the revised manuscript, page 23, line 5-10.*

5. *“In Figure 5E-G, Jurkat cells migrate towards tumor regions, even in the absence of IFN-g, despite the negligible upregulation of T-cell chemokines like CXCL10 by treatment with Birinapant alone in Figure 4. If they use Jurkat cells as the immune cells to supply IFN-g, cancer and immune cells are separately cultured in this system which is contrary to the co-culture system in Figure 2, and activation of Jurkat cells to secrete IFN-g remains unclear. To confirm their model, they should check the expression of IFN-g in Jurkat cells in the presence or absence of cancer cells. Probably they would be able to purify Jurkat cells after the co-culture experiment by a cell sorter using red color and examine RT-qPCR or another way.”*

Response: In Figure 5E-G, Jurkat cells were activated with IL2 and anti-CD3 antibody in the co-culture assay. We found that IFN- γ mRNA level of Jurkat cells significantly increased upon activation by IL2 and anti-CD3 antibody even in the absence of cancer cells (Fig. S10C). Therefore, the activated Jurkat cells were used as the immune cells to supply IFN- γ , which can synergize with IAPi to activate STING and T-cell chemokines like CXCL10 and thus promote immune cell migration. Similarly, IL2 and anti-CD3 antibody were used to pre-activate immune cells in the co-culture assay throughout the studies. See revised Fig. S10C, page 14, line 1-2.

6. *“STAT1 is a representative downstream of IFN-g signaling, and its expression should be increased following IFN-g treatment. Unlike STAT3, TBK1, and IRF3, whose activation is mainly regulated by phosphorylation alone, it is well-known that STAT1 is regulated at the expression level in addition to phosphorylation. Indeed, transcriptome analysis conducted by the authors also included STAT1 among the up-regulated genes following Birinapant treatment in Figure 3A. Conversely, in their western blotting, STAT1 expression did not show significant change, which is like internal control, following the treatment with Birinapant, and even in the presence of IFN-g for 24 hours in Figure 7G. This discrepancy should be explained/reconciled.”*

Response: We have corrected the typo from 24 hours to 1 hour, where we did not observe STAT protein level change upon short time treatment. See page 45, line 22.

7. *“In Figure S7, the mechanism responsible for the accumulation of cytoplasmic DNA following the administration of an IAP1 inhibitor or IFN-g remains totally unclear. If it is true, I think this phenomenon could be crucial and informative in this field, suggesting cGAS activation following IAP1 inhibitor or IFN-g administration. They should check the concentration of cytosolic 2’3’ cGAMP by ELISA or another technique in the presence of IAP1 inhibitors or IFN-g to confirm their findings.”*

Response: We have checked the cytosolic 2'3' cGAMP by ELISA and found a significant increase of cytosolic 2'3' cGAMP level in the presence of an IAPi birinapant and IFN-g. See revised Fig. S8C, page 13, line 11-12.

Reviewer #3 (Remarks to the Author): with expertise in lung cancer, LKB1, immunotherapy

The authors use a variety of assays their group has developed over the years including screening for onco-immune protein-protein interaction (PPI), screening of chemical libraries with a high-throughput immunomodulator phenotype (HTiP) assay for chemicals that enhance killing of human tumor cells by human peripheral blood mononuclear cells to look for targets for treating LKB1 mutant lung adenocarcinomas (LUAD). Their primary preclinical model are human LUAD cell lines. They discover that cIAP1 interacts with LKB1 and that known inhibitors of cIAP1 (SMAC mimetics) block this interaction to increase the expression of STING in tumor cells and enhance immune cell killing and immune cell migration in the mixed culture in vitro assay. They also work out the domains of LKB1 responsible for the cIAP1 interaction and demonstrate that JAK1-STAT1 mediates the IAP inhibitor effect in LKB1 mutant LUAD. They emphasize that LKB1 loss of function in LUADs leads to an "IAP dependency." In a syngeneic mouse model they show an cIAP1 inhibitor, birinapant, inhibits tumor cell growth and this is dependent on having an intact immune system. They conclude: "Onco-Immune Protein-Protein Interacton (OI-PPI) mapping reveals a rewired LKB1-IAP-JAK dynamic complex, informing IAP-dependency and vulnerability of LKB1-mutant tumors. Targeting this rewired OI-PPI induces immune-dependent suppression of LKB1-mutant tumors in preclinical in vitro and in vivo models, demonstrating its therapeutic potential to accelerate oncogenic alteration-directed precision immunotherapy". No data are presented: on expression of the various factors they discovered in preclinical models in patient LKB1 mutant tumors; and no data are presented on the impact of immune checkpoint combined with IAP inhibitor therapy.

Comments to the authors:

This manuscript addresses a major knowledge gap – namely what is the mechanism underlying the "immunologically cold phenotype" of LKB1/STK11 mutant LUADs and how could this be therapeutically targeted. The paper is reviewed in the context of other recent multiple, high profile papers, addressing this same topic.

1. All of the experiments are technically well done a clearly presented.
2. Their discoveries of LKB1- cIAP1 interaction, and the subsequent JAK1-STAT1 pathway and STING effects are important and new. These clearly add to the field as a whole and fill in parts of the knowledge gap.
3. The discovery of the role of SMAC mimetics as potential therapeutics has immediate clinical translational relevance. This is also important given the initial high expectations for SMAC mimetics as cancer therapeutics yet their poor performance in the clinic to date.

4. “As presented a major limitation of the current study is the information on the *in vivo* efficacy of the proposed treatment using only one mouse syngeneic model (WRJ388). First, this model derived from a genetically engineered mouse model (GEMM) by the authors (see their reference #36) is KRAS mutant/LKB1 mutant but TP53 wildtype. There are several other GEMM LUAD models that are KRAS, TP53, and LKB1 mutant used by other investigators. Given the frequent co-occurrence of KRAS, TP53 and LKB1 mutations before any clinical translation we will need to know the impact of a TP53 mutation on therapeutic targeting with SMAC mimetics. Second, as the authors describe the birinapant treatment led to a “~42% reduction in tumor volume..”. While this is statistically significant, what we are looking for to put the effort into tough clinical trials, are long term control/cures of LKB1 mutant LUADs in xenograft and syngeneic mouse models. The time period (12 days) of their *in vivo* experiment was very short compared to most of similar types of experiments reported in the literature which usually extend over multiple weeks. Finally, what we all want to know is the impact of available immune checkpoint blockade (ICB) added to this system. Could the combination of a SMAC mimetic plus ICB give long term control/potential cures in this syngeneic mouse model? It could be, that they do, or just as important they may not – whatever the results are we need to know. More information related to *in vivo* efficacy and/or the role of ICB, and on models with a TP53 mutation, would have greatly enhanced the value of this manuscript.”

Response: We selected the WRJ388 and newly generated CMT167 mouse model because of their relevance to LUAD patients, where a major patient subpopulation was found to harbor KRAS mutant/LKB1 mutant but TP53 wildtype. There are only a few cases of patients with co-occurring KRAS, TP53 and LKB1 mutations found in TCGA cohort.

To test the combination effect of IAP inhibitors and ICIs, we first examined the immune response of WRJ388 to an ICI, *InVivoMAb* anti-mouse PD-1 (CD279, BioXcell). We found that WRJ388 cells respond well to ICI at 200 µg per mouse, which gave a narrow window for testing the combination effect. To address reviewer’s comment for testing the combination effect, we developed a CMT167 cell based syngeneic model that are resistant to anti-PD-1 treatment at 200 µg dosage. We found that the LKB1-KO CMT167 cells showed significantly stronger *in vivo* tumorigenicity capacity than the LKB1-WT control in the immune-competent mice (Fig. S11G). Moreover, the LKB1-KO CMT167 cells showed resistance to anti-PD1 (200 µg/mouse) immunotherapy treatment (Fig. S11G-H). While treatment of an IAPi AT406 alone slightly slowed down the LKB1-KO CMT167 tumor growth as compared to the DMSO control, the combination of AT406 and anti-PD1 led to a significant synergistic anti-tumor effect (Fig. S11H). Further, depletion of CD8+, but not CD4+, T cells significantly abolished the anti-tumor activity of the AT406 and anti-PD1 combination treatment (Fig. S11I), suggesting the underlying immune killing effect and CD8+ T cell-dependency. See revised Fig. S11G-H, page 16 (line 19-23) and page 17 (line 1-7).

We acknowledge the importance of long-term survival studies to comprehensively evaluate the *in vivo* efficacy of IAP inhibitor and its combination effect with anti-PD1 immunotherapy on the treatment of lung cancer patients. We focused our initial studies on the *in vivo* efficacy of IAP inhibitors through the short-term treatment to gain mechanistic insights into the LKB1-IAP-JAK regulatory complex in LKB1 mutated tumors.

This mechanistic study has set the stage for future long-term investigations as recommended by the reviewer, which is important for clinical translation. Accordingly, we have discussed limitations of the current study (See revised discussion, page 23, line 17-23).

5. *“The discussion lacked two major things; any kind of “limitations of the current study” discussion; and also a discussion by the authors placing their study in the context and their view of its integration with all of the other recent papers which have identified potential therapeutic vulnerabilities in LKB1 mutant LUAD preclinical model systems (human and mouse models).”*

Response: We have added discussion on limitations of the current study and other papers that identified potential therapeutic vulnerabilities in LKB1 mutant LUAD model systems. See page 20, line 4-10, and page 23, line 17-23.

6. *“Their findings clearly have implications for what should be found in studies LKB1 mutant LUAD in patient tumor specimens with regard to the JAK1-STAT1, Sting pathways. Data on this from immuno-histochemical or multi-omics studies would have enhanced the value of the paper, whatever the results were.”*

Response: We have performed multi-omics studies using TCGA patient data. We found that a panel of signature genes related to JAK-STAT and STING pathway were significantly downregulated in LKB1-mut LUAD samples (Fig. S5C), suggesting the clinical relevance of our findings. See revised Fig. S5C, page 11, line 7-8.

7. *“Finally, I found it interesting and amusing to read the bioRxiv preprint version of this paper uploaded in 2021 (bioRxiv preprint doi: <https://doi.org/10.1101/2021.09.17.460294>). Clearly, the authors made the discovery in an entirely different order than the presented in the current version of the manuscript – they performed the chemical screen for therapeutics which led to the discovery of the SMAC mimetic effect in their in vitro co-culture system and that subsequently led them to look for the protein interaction. In many respects, I found their bioRxiv paper and what probably actually happened to be more scientific discovery based “compelling” (I could see how they made their discovery) than the current description of the course of events. My guess is many other readers interested in this topic will note the same thing. “What happens in bioRxivs – does not stay (“hidden”) in bioRxivs.”*

Response: All the results in bioRxiv version are included in the current version. The chemical screen and PPI screen were done in parallel. Results from these two unbiased profiling studies converged at the LKB1-IAP connectivity. The current version focuses on the molecular connectivity of the tumor intrinsic factors with the immune response pathways, emphasizing the mechanistic basis of mutated LKB1-invoked immune resistance. The bioRxiv manuscript as initially uploaded focused on the pharmacological effect of the IAPi on restoring the immune response of LKB1-mutant lung cancer cells. Upon consideration, we decided to focus on mechanistic investigations for the formal submission.

Response to reviewers' comments

Shu *et al*, Uncovering the rewired IAP-JAK regulatory axis as an immune-dependent vulnerability of LKB1-mutant lung cancer

REVIEWERS' COMMENTS

Reviewer #1 (Remarks to the Author):

The authors have satisfactorily addressed all of my concerns. In addition, they have greatly strengthened the manuscript by including additional requested information and data in the supplemental tables or in revised and new figures. This includes new experiments that (1) prove their main conclusion using a second LKB1-mutant cell line, (2) demonstrate efficacy by combining a Smac mimetic with an anti-PD1 biologic in their model, (3) demonstrate the IAP selectivity for association with JAK1 to also include the cIAP1 similar IAP, cIAP2, but not the X-linked IAP, XIAP, and (4) the demonstrate (possibly for the first time) that JAK1 can phosphorylate the associated cIAP1 protein.

Response:

Thanks to Reviewer #1 for your positive comments.

Reviewer #2 (Remarks to the Author):

The authors provided enough data to convince the comments, making the paper much more informative. Their manuscript can be accepted.

Response:

Thanks to Reviewer #2 for your positive comments.

Reviewer #3 (Remarks to the Author):

The authors submit a revised version of their manuscript with a 13 page rebuttal section addressing concerns of the 3 reviewers. The rebuttal includes additional experimental data and the manuscript has been edited to address the various concerns and the new added information. I am addressing my concerns as Reviewer #3 to their response to my major comment which I have included here for easy reference:

“4. “As presented a major limitation of the current study is the information on the in vivo efficacy of the proposed treatment using only one mouse syngeneic model (WRJ388). First, this model derived from a genetically engineered mouse model (GEMM) by the authors (see their reference #36) is KRAS mutant/LKB1 mutant but TP53 wildtype. There are several other GEMM LUAD models that are KRAS, TP53, and LKB1 mutant used by

other investigators. Given the frequent co-occurrence of KRAS, TP53 and LKB1 mutations before any clinical translation we will need to know the impact of a TP53 mutation on therapeutic targeting with SMAC mimetics. Second, as the authors describe the birinapant treatment led to a “~42% reduction in tumor volume..”. While this is statistically significant, what we are looking for to put the effort into tough clinical trials, are long term control/cures of LKB1 mutant LUADs in xenograft and syngeneic mouse models. The time period (12 days) of their in vivo experiment was very short compared to most of similar types of experiments reported in the literature which usually extend over multiple weeks. Finally, what we all want to know is the impact of available immune checkpoint blockade (ICB) added to this system. Could the combination of a SMAC mimetic plus ICB give long term control/potential cures in this syngeneic mouse model? It could be, that they do, or just as important they may not – whatever the results are we need to know. More information related to in vivo efficacy and/or the role of ICB, and on models with a TP53 mutation, would have greatly enhanced the value of this manuscript.”

The major point was that the real ultimate translational value of their work was whether there were quantitatively large in vivo effects of combined targeting using SMAC mimetic and ICB. Another key point dealt with potential testing of their overall hypothesis using GEMM derived syngeneic mouse models of mutant KRAS, TP53, with and without LKB1/STK11 inactivation (“KP”, “KPL” lines) that have been widely used for similar studies. Unfortunately they choose not to study this model, and also essentially “wrote off” any role of TP53 abnormalities. For purposes of this review, I will put aside any consideration of there not using a widely available model, and not determining the role of TP53. They did choose to study CMT167 mouse cells, which caused me to go to their methods and references to find out about these cells, what were their characteristics, and where did they get them. I spent a good deal of time searching their manuscript and figure legends but to know avail. At that point I went to the literature and found out about these cells (the parent line CMT64 derived in the 1970’s as metastasizing mouse lung cancer (FRANKS, L.M., CARBONELL, A.W., HEMMING, V.J. & RIDDLE, P.N. (1976). Metastasizing tumours from serum-supplemented and serum-free cell lines from a C57BL mouse lung tumour. *Cancer Res.*, 36, 1049.” and characterized in 1984 with information on the metastatic variants (Br. J. Cancer (1984), 49, 415-421 Heterogeneity in a spontaneous mouse lung carcinoma: selection and characterisation of stable metastatic variants M.G. Layton & L.M. Franks”. It has been used in several publications from the Nemenoff lab who made the interesting observation that response to ICB was dramatically different in orthotopic vs. subcutaneous models who cited his source, and the derivation of the cell line and also confirmed that it had a KRASG12V mutation previously reported by Justilien et al. The status of TP53 and LKB1 are unknown. (Li et al. The Tumor Microenvironment Regulates Sensitivity of Murine Lung Tumors to PD-1/PD-L1 Antibody Blockade, *Cancer Immunol Res*; 5(9) September 2017, 767-777; Justilien V, Regala RP, Tseng I-C, Walsh MP, Batra J, et al. (2012) Matrix Metalloproteinase-10 Is Required for Lung Cancer Stem Cell Maintenance, Tumor Initiation and Metastatic Potential. *PLoS ONE* 7(4): e35040. doi:10.1371/journal.pone.0035040). I note the Nemenoff studies went for 3 weeks. The part was the length of time of studying the tumor responses to various treatment being very short – for the WRJ388 it is 12 days for CMT167 it is 9 days.

I am pointing these things out, because if this paper was presented to our grad student/post doc journal club, all of these issues would have come up. While a lot of fancy mechanism stuff is fine, what really counts is –“does this really work? Is it really worth pursuing clinically?” Of course, it is essential that the methods section document this key reagent and also that the literature is properly cited. In addition, while they have a “limitation of the current study” ending their discussion – it is pretty “tepid” given the real work that still needs to be done.

Response:

We have addressed Reviewer 3's comments by expanding the biological and clinical relevance of our findings as a potential therapeutic approach, detailing cell resources, adding relevant references, and acknowledging the study's limitations and model constraints.

We have added detailed information and relevant references of the CMT167 models in the Method section. Specifically, we added that “CMT167 cells (CancerTools, Cat. #: 151448) were cultured in DMEM medium. CMT167 cells were derived from metastasizing mouse lung cancer CMT64 cells and carry KRAS^{G12V} mutation, and are TP53- and LKB1-WT⁹⁰⁻⁹³.”

We have discussed the biological and clinical relevance of our findings in the Discussion section. Specifically, we have discussed that “To further determine the translational potential of our findings, studies with long-term survival experiments in more disease-relevant GEMM models are needed. In our *in vivo* study, the mice in control group reached the maximal tumour burden permitted in a short-term of two weeks. Therefore, we evaluated the *in vivo* efficacy of IAP inhibitors with the same short-term treatment, and collected paired tumor samples to gain mechanistic insights into the LKB1-IAP-JAK regulatory complex in LKB1 mutated tumors. However, we recognize the importance of long-term survival studies to comprehensively evaluate the *in vivo* efficacy of IAP inhibitors and its combination effect with anti-PD1 immunotherapy for clinical translation. In addition, we used subcutaneous syngeneic mouse models, which may have distinct immunological profiles as compared to the orthotopic GEMM models. Future studies using additional LKB1-mut LUAD GEMM models by considering the clinically relevant co-occurring mutated genes, such as *KRAS* and *TP53*, are needed to delve deeper into the long-term effect and to determine potential treatment-associated adaptive resistance mechanisms.” See the revised manuscript (page 23 and page 24).